# Loss of full-length dystrophin expression results in major cell-autonomous abnormalities in proliferating myoblasts

Maxime RF Gosselin[1], Virginie Mournetas[2], Malgorzata Borczyk[3], Suraj Verma[4], Annalisa Occhipinti[4], Justyna Róg[1,5], Lukasz Bozycki[1,5], Michal Korostynski[3], Samuel C Robson[1,6], Claudio Angione[4], Christian Pinset[7], Dariusz C Gorecki[1]*

[1]School of Pharmacy and Biomedical Sciences, University of Portsmouth, Portsmouth, United Kingdom; [2]INSERM UEVE UMR861, I-STEM, AFM, Corbeil-Essonnes, France; [3]Laboratory of Pharmacogenomics, Maj Institute of Pharmacology PAS, Krakow, Poland; [4]School of Computing, Engineering and Digital Technologies, Teesside University, Middlesbrough, United Kingdom; [5]Laboratory of Cellular Metabolism, Nencki Institute of Experimental Biology, Warsaw, Poland; [6]Centre for Enzyme Innovation, University of Portsmouth, Portsmouth, United Kingdom; [7]CNRS, I-STEM, AFM, Corbeil-Essonnes, France

*For correspondence:
darek.gorecki@port.ac.uk

**Abstract** Duchenne muscular dystrophy (DMD) affects myofibers and muscle stem cells, causing progressive muscle degeneration and repair defects. It was unknown whether dystrophic myoblasts—the effector cells of muscle growth and regeneration—are affected. Using transcriptomic, genome-scale metabolic modelling and functional analyses, we demonstrate, for the first time, convergent abnormalities in primary mouse and human dystrophic myoblasts. In *Dmd*[mdx] myoblasts lacking full-length dystrophin, the expression of 170 genes was significantly altered. *Myod1* and key genes controlled by MyoD (*Myog*, *Mymk*, *Mymx*, epigenetic regulators, ECM interactors, calcium signalling and fibrosis genes) were significantly downregulated. Gene ontology analysis indicated enrichment in genes involved in muscle development and function. Functionally, we found increased myoblast proliferation, reduced chemotaxis and accelerated differentiation, which are all essential for myoregeneration. The defects were caused by the loss of expression of full-length dystrophin, as similar and not exacerbated alterations were observed in dystrophin-null *Dmd*[mdx-βgeo] myoblasts. Corresponding abnormalities were identified in human DMD primary myoblasts and a dystrophic mouse muscle cell line, confirming the cross-species and cell-autonomous nature of these defects. The genome-scale metabolic analysis in human DMD myoblasts showed alterations in the rate of glycolysis/gluconeogenesis, leukotriene metabolism, and mitochondrial beta-oxidation of various fatty acids. These results reveal the disease continuum: DMD defects in satellite cells, the myoblast dysfunction affecting muscle regeneration, which is insufficient to counteract muscle loss due to myofiber instability. Contrary to the established belief, our data demonstrate that DMD abnormalities occur in myoblasts, making these cells a novel therapeutic target for the treatment of this lethal disease.

## Editor's evaluation

This is an in depth analysis of the transcriptomic changes occurring in mouse and human myogenic cells that lack of dystrophin. The alterations have implications for the pathogenesis of Duchenne muscular dystrophy. The findings could lead to new therapeutic interventions directed at this highly disabling and lethal disease.

## Introduction

Duchenne muscular dystrophy (DMD) is a debilitating and lethal neuromuscular disorder caused by mutations in the *DMD* gene located on the X chromosome (*Hoffman et al., 1988*). Diagnosis is made between the age of 2 and 5, loss of ambulation occurs around 12 and young adults die due to respiratory and/or cardiac failure (*Koeks et al., 2017*).

*DMD* is the largest human gene known (*Tennyson et al., 1995*). Three full-length transcripts encode 427 kDa proteins, while further intragenic promoters drive the expression of progressively truncated variants. Dp427 and the dystrophin-associated protein complex (DAPC) are important for the functional development of differentiating myotubes (*Shoji et al., 2015*) and subsequently prevent contraction-induced injury in the mature muscle (*Rader et al., 2016*).

Several studies showed that the ablation of dystrophin in fully differentiated myofibres did not trigger their degeneration (*Rader et al., 2016*; *Ghahramani Seno et al., 2008*), and even that myofibres can function without dystrophin (*Waugh et al., 2014*; *Vieira et al., 2015*).

In fact, DMD pathology is active prior to diagnosis: delays in the attainment of motor and non-motor milestones are discernible in 2 months old DMD babies (*van Dommelen et al., 2020*) and transcriptomes of muscles from asymptomatic DMD patients revealed typical dystrophic abnormalities (*Pescatori et al., 2007*). Studies of human foetuses (*Toop and Emery, 1977*; *Emery, 1977*; *Vassilopoulos and Emery, 1977*) and various animal DMD models, including GRMD dogs (*Nguyen et al., 2002*), Sapje zebrafish (*Bassett et al., 1977*) and *Dmd*[mdx] mouse embryos (*Merrick et al., 2009*) revealed that the pathology starts already in prenatal development.

Indeed, in skeletal muscle lineages modelled in human DMD pluripotent stem cells, we have recently demonstrated marked transcriptome and miRNA dysregulations identifiable even before muscle specialisation (*Mournetas et al., 2021*). These data, combined with the existence of the specific embryonic dystrophin Dp412e (*Massouridès et al., 2015*), substantiate the early disease manifestations in muscle precursor cells.

Importantly, there is increasing evidence that dysregulation of myogenic cells is behind muscle pathology and disease progression in adult DMD muscle. However, the mechanism(s) remain(s) to be elucidated and, given the interaction of myogenic and inflammatory cells in muscle regeneration (*Tidball, 2017*), distinguishing the primary and the secondary consequences of dystrophin deficiency is crucial for the development of effective therapies.

Helen Blau proposed that DMD is intrinsic to the undifferentiated myoblast (*Blau et al., 1983*). This hypothesis was initially discounted (*Hurko et al., 1987*), but there is new evidence that *DMD* mutations produce a range of cell-autonomous abnormalities in both human and mouse myogenic cells from adult muscles (*Yablonka-Reuveni and Anderson, 2006*; *Yeung et al., 2006*; *Sacco et al., 2010*; *Górecki, 2016*).

If DMD directly affects myoblasts, cells that are key to muscle regeneration, a better definition of the consequences of the loss of *DMD* gene expression could help identify early disease biomarkers and establish better therapeutic targets.

Herein, using a combination of RNA-Seq, molecular and functional approaches, we compared dystrophic and healthy myoblasts isolated from skeletal muscles of the commonly used *Dmd*[mdx] mouse model. The *Dmd*[mdx] has a stop codon mutation in exon 23 (*Sicinski et al., 1989*) resulting in the loss of full-length (Dp427) dystrophin expression, which reflects the molecular defect affecting the majority of DMD patients. Detailed histological and molecular analyses indicated that the clinical phenotype in 3–10 weeks old *Dmd*[mdx] is as aggressive as in Duchenne patients (*Duddy et al., 2015*; *DiMario et al., 1991*; *Massopust et al., 2020*). Therefore, during this period, *Dmd*[mdx] is a good model for the human pathology. Unlike human, after this phase of severe degeneration-regeneration, most *Dmd*[mdx] muscles start to regenerate (*DiMario et al., 1991*) due to an unknown mechanism.

While Dp427 is expressed in satellite cells and myofibres and its loss in *Dmd*[mdx] reflects the predominant molecular defect in DMD patients, dystrophin Dp71 is present in regenerating muscles and specifically in myoblasts (*Howard et al., 1999*). Yet, little attention has been given to the DMD pathology in the dystrophin-null individuals. Given that we have recently described exacerbated pathology in *Dmd*[mdx-βgeo] mice lacking all dystrophins (*Young et al., 2020*), we also investigated the consequences of the total loss of *Dmd* expression in *Dmd*[mdx-βgeo] myoblasts (*Wertz and Füchtbauer, 1998*).

Finally, we compared transcriptomic alterations in mouse and human dystrophic myoblasts to identify defects occurring across species. Given that skeletal muscle is a key metabolic site, accounting for

around 90% of total oxygen uptake, muscle metabolism is a significant variable (*Zurlo et al., 1990*). Moreover, metabolic dysfunction can impair muscle regeneration, and the impact of DMD on cellular energetics in muscle progenitors has recently been identified (*Matre et al., 2019*). Therefore, using the RNA-Seq profiles, we reconstructed the human myoblast-specific enzyme-constrained genome-scale metabolic model.

We report that the absence of expression of the full-length dystrophin triggers major transcriptomic and functional abnormalities in myoblasts. These abnormalities are cell autonomous, as they persist in myogenic cells maintained long-term in culture. Importantly, key alterations are common between mouse and human myoblasts.

## Results

### Transcriptomic alterations in proliferating *Dmd*^mdx^ myoblasts

Total RNA extracted from primary myoblasts isolated from gastrocnemii of 8-week-old male *Dmd*^mdx^, *Dmd*^mdx-βgeo^ and control mice was subjected to RNA-Seq and analysed for the differential expression of genes between groups and the enrichment of GO categories in the generated lists of differentially expressed genes.

Bioinformatic analysis of the RNA-Seq data showed the impact of the *Dmd*^mdx^ allele on the primary myoblast transcriptome. Sample-based hierarchical clustering clearly segregates genes into two groups corresponding to genotypes (*Figure 1a*) and a volcano plot illustrates the presence of a substantial number of significantly dysregulated genes (*Figure 1b*, *Supplementary file 1-1*).

The expression of 170 genes was found to be significantly up- or down-regulated more than two-fold when comparing *Dmd*^mdx^ and WT primary mouse myoblasts (*Supplementary file 1-1*). Among these, *Myod1* and *Myog,* key coordinators of skeletal muscle development and repair, were found to be downregulated in dystrophic myoblasts and their downregulation was confirmed by qPCR (*Figure 1—figure supplement 1*). *Pax3* and *Pax7* transcriptional regulators (*Magli et al., 2019*) were also individually investigated, with Pax3 levels confirmed as significantly downregulated by qPCR (*Figure 1—figure supplement 1*). Our analysis identified downregulation of further important regulators and effectors of the muscle program (*Figure 1c*, *Supplementary file 1*) such as *Mymx and Bex1*, known to be regulators of muscle repair (*Koo et al., 2007*). *Des*, a structural component of myofilaments, and *Itga7*, the primary laminin-1 receptor of myoblasts and mature fibres were also found to be significantly downregulated (*Figure 1c*, *Supplementary file 1*).

The *H19* gene, defined as promoting differentiation (*Qin et al., 2017*), was found to be significantly downregulated in proliferating *Dmd*^mdx^ myoblasts (*Figure 1c*, *Supplementary file 1*), which combined with reduced expression of aforementioned muscle program markers, as well as myogenic differentiation markers such as *Acta1*, *Actc1*, *Actn3*, *Atp2a1* and upregulation of *Nov,* known to inhibit myogenic differentiation (*Sakamoto et al., 2002*), imply an altered readiness of dystrophic myoblasts to differentiate. Moreover, numerous genes encoding histones and *Smyd1* (*Li et al., 2009b*) and *Hmga1* (*Harrer et al., 2004*; *Qiu et al., 2017*) regulators of chromatin organisation/chromatin interacting proteins, were found to be significantly dysregulated in dystrophic myoblasts (*Figure 1c*, *Supplementary file 1*). Taken together, these gene expression changes indicated an increased readiness of dystrophic myoblasts to proliferate, compared to their WT counterparts.

Several genes key to myoblast-extracellular matrix interactions and fibrosis were found to be significantly dysregulated in *Dmd*^mdx^ myoblasts (*Figure 1c*, *Supplementary file 1*), including: *Mmp3*, *Mmp9* and *Mmp10* metalloproteases, collagen genes (*Col2a1*, *Col4a1*, *Col4a2,* and *Col8a2*), *Lrrn1* a transmembrane protein interacting with fibronectin (*Haines et al., 2005*) as well as genes encoding fibronectin itself (*Fn1*) and decorin (*Dcn*). The latter promotes proliferation and differentiation in myoblasts (*Kishioka et al., 2008*) but also has a role in collagen assembly and mineralisation (*Mochida et al., 2009*).

Finally, calsequestrins (*Casq1* and *Casq2*) (*Supplementary file 1-1*) were significantly downregulated, in line with previous observations in the dystrophic muscle (*Doran et al., 2004*).

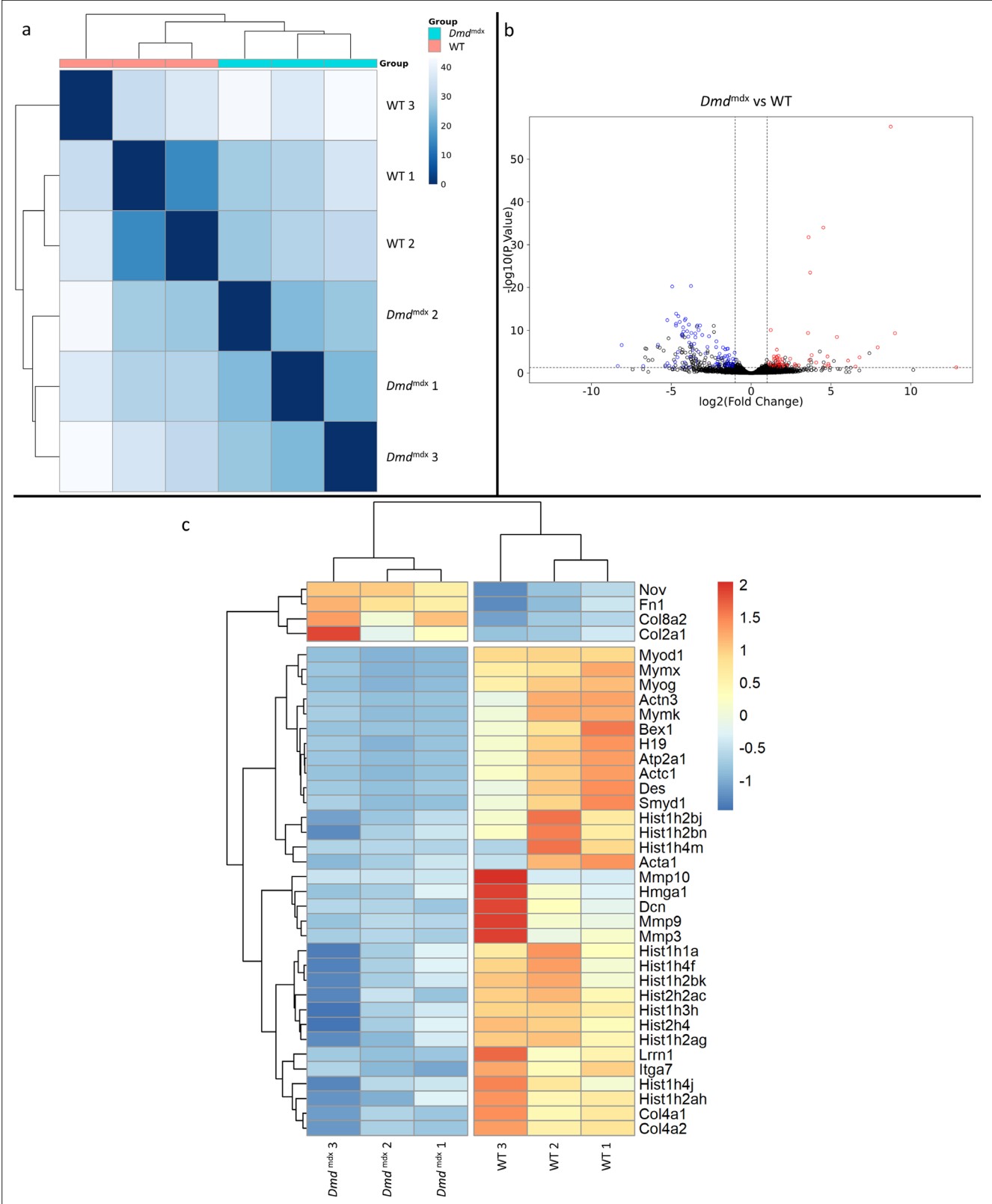

**Figure 1.** Differential gene expression between *Dmd*^mdx and WT mouse primary myoblasts. (**a**) Sample-based hierarchical clustering heatmap based on the top 500 genes with the highest standard deviation for *Dmd*^mdx vs. WT based on RNA-Seq data. Colour represents dissimilarity between samples, based on the Euclidean distance, from dark blue (0) for identical samples to white (40) for the most distinct. (**b**) Volcano plot for *Dmd*^mdx versus WT primary mouse myoblast differential gene expression analysis. Circles represent individual genes, with colour representing significance and direction

*Figure 1 continued on next page*

*Figure 1 continued*

of dysregulation (blue-significantly downregulated; red-significantly upregulated; black-not significantly dysregulated). Circle position represents fold change on a log2 scale for the x-axis and adjusted p-value on a -log10 scale for the y-axis. Horizontal dotted line represents the threshold of an adjusted p-value of 5.0e-2 or lower, while the vertical dotted lines represent the threshold for the absolute log2 fold change of 1. (**c**) Clustered heatmap of genes of interest from the dysregulated gene list of *Dmd*^mdx vs. WT primary myoblasts. Colour represents the z-score such that each gene has a mean of 0 and standard deviation of 1 to allow direct comparisons.

The online version of this article includes the following figure supplement(s) for figure 1:

**Figure supplement 1.** Differential expression of muscle programme markers in primary myoblasts from *Dmd*^mdx and *Dmd*^mdx-βgeo mice.

## No exacerbation of transcriptomic alterations in the dystrophin-null *Dmd*^mdx-βgeo compared to *Dmd*^mdx myoblasts

Corresponding bioinformatic comparison of *Dmd*^mdx-βgeo vs. *Dmd*^mdx myoblast transcriptomes found no substantial difference between them, as illustrated by sample clustering (*Figure 2a*) and volcano plot (*Figure 2b*), with no segregation between groups. Only 11 genes were found to be significantly differentially expressed between the two genotypes. Of these, 6 are pseudogenes, 1 is a processed transcript and only 4 are protein-coding genes (*Igf1*, *Npr3*, *Postn* and *Capn6*). None of these genes has a higher log2 fold change than 1.7 and the adjusted *p*-values have an average of 1.0e-2 compared to 9.6e$^{-3}$ for the genes found to be significantly differentially expressed between *Dmd*^mdx and WT myoblasts.

Interestingly, a comparison of the *Dmd*^mdx-βgeo to the WT transcriptome returned fewer (81) (*Supplementary file 1-2*) significantly differentially expressed genes than *Dmd*^mdx versus WT (170) (*Supplementary file 1-1*). However, a comparison of the fold changes between the two analyses (*Figure 2c*) revealed that the two models show strikingly similar alterations with a strong, significant correlation (*r*=0.9418, p<1.0e-4) in the log2 fold changes for genes significantly altered in *Dmd*^mdx versus WT and/ or *Dmd*^mdx-βgeo versus WT analyses (*Figure 2c*).

## MyoD-dependent downregulation of gene expression in dystrophic myoblasts

Lists of genes altered in dystrophic myoblasts were analysed for common patterns of transcriptional regulation. Overrepresented transcription factor binding sites (TFBS) on promoter regions of the dysregulated genes were examined using the seqinspector tool and available ChIP-Seq data (*ENCODE Project Consortium, 2004*). Genes found downregulated in *Dmd*^mdx had increased ChIP-Seq signal at the TFBS for four transcription factors: MyoD (adjusted *p*-value: 2.9e$^{-21}$, track GEO accession GSM915165), TCF12 (1e$^{-19}$, GSM915178), MYOG (5.6e$^{-19}$, GSM915164), and TCF3 (adjusted *p*-value 3.3e-3, GSM915177) (*Figure 3*, *Supplementary file 2B*). A range of genes, including *Mymk*, *Mymx*, *Chrna1*, and *Acta1* showed a binding signal at their TFBS above the background of at least three out of four of these overrepresented TFs (*Figure 3b*, *Supplementary file 2C*). Furthermore, transcripts downregulated in *Dmd*^mdx-βgeo also exhibit overrepresentation of MyoD binding sites (*Supplementary file 2E*). In contrast, genes upregulated in any cell type under investigation did not show any statistically significant overrepresentation of the binding signal of any TF (*Supplementary file 2A and D*).

## GO enrichment analysis indicates significant functional alterations in dystrophic myoblasts

Two broad GO categories: 'muscle system process' and 'regulation of muscle system process' were enriched in the downregulated genes found when comparing *Dmd*^mdx and *Dmd*^mdx-βgeo vs. WT primary mouse myoblasts (*Figure 4a*, *Supplementary file 1-2*). These categories include genes involved in muscle development and function. For a myoblast to fulfil its role in regenerating a damaged muscle, it must be able to proliferate, migrate towards and differentiate in response to relevant stimuli. GO enrichment analyses of downregulated genes (*Figure 4*, *Supplementary file 1-3*) imply phenotypic alterations in dystrophic myoblasts affecting all three functions.

Specifically, categories such as 'chromatin assembly' and 'muscle cell proliferation' (*Figure 4b*, *Supplementary file 1-3*), are enriched, indicating an altered proliferative state of dystrophic myoblasts. The enrichment of the GO categories 'muscle cell migration' and 'positive regulation of cell migration' in downregulated genes (*Figure 4*, *Supplementary file 1-3*) indicate a migrative phenotype in

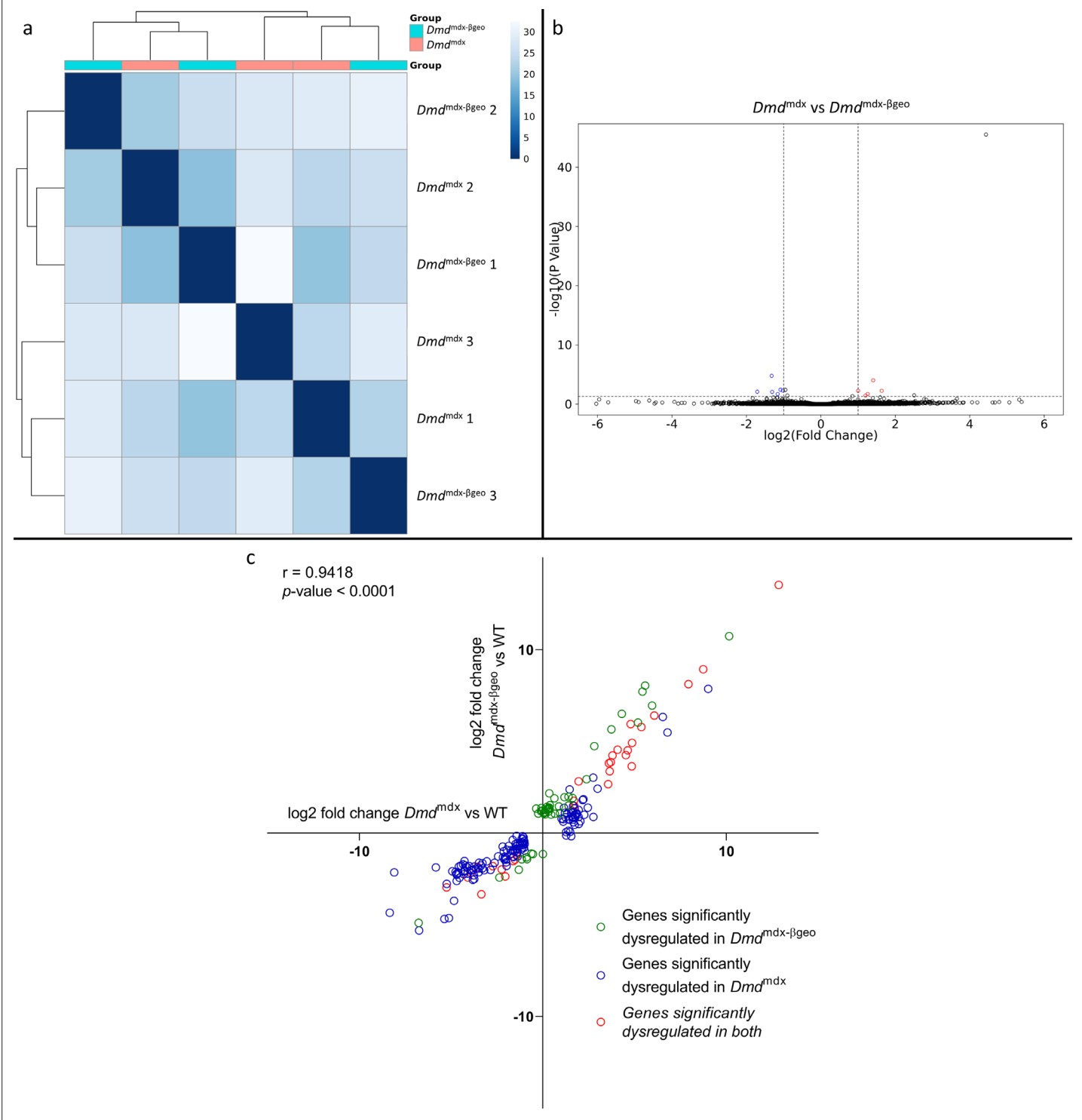

**Figure 2.** Correlation of significantly dysregulated gene expressions between *Dmd*^mdx and *Dmd*^mdx-βgeo vs WT. (**a**) Sample-based hierarchical clustering heatmap based on the top 500 genes with the highest standard deviation for *Dmd*^mdx-βgeo vs. *Dmd*^mdx Colour represents dissimilarity between samples based on the Euclidean distance, from dark blue (0) for identical samples to white (40) for the most distinct. Note the absence of sample segregation into two groups corresponding to genotypes. (**b**) Volcano plot for *Dmd*^mdx-βgeo vs. *Dmd*^mdx primary mouse myoblast differential gene expression analysis. Circles represent individual genes, with colour representing significance and direction of dysregulation (blue-significantly downregulated; red-significantly upregulated; black-not significantly dysregulated). Circle position represents fold change on a log2 scale for the x-axis and adjusted *p*-value on a -log10 scale for the y-axis. Horizontal dotted line represents the threshold of an adjusted *p*-value of 5.0e-2 or lower while the vertical doted lines represent the threshold for the absolute log2 fold change of 1. (**c**) log2 fold change values (*Dmd*^mdx versus WT on the x-axis and *Dmd*^mdx-βgeo versus WT

*Figure 2 continued on next page*

*Figure 2 continued*

on the y-axis) for genes significantly dysregulated in one or both dystrophic primary mouse myoblasts vs. WT. Pearson's correlation coefficients (**r**) and *p*-value are shown. Green circles represent genes significantly dysregulated in *Dmd*<sup>mdx-βgeo</sup> vs. WT, blue circles represent genes significantly dysregulated in *Dmd*<sup>mdx</sup> vs. WT and red circles represent gernes significantly dysregulated in both.

dystrophic myoblasts, while GO categories 'muscle cell differentiation' and 'striated muscle cell differentiation' (*Figure 4a and b- Supplementary files 1-3*) suggest that dystrophic myoblasts may have an altered ability to differentiate.

Moreover, several GO categories related to calcium ion transport, homeostasis and calcium-mediated signalling were found significantly enriched in *Dmd*<sup>mdx</sup> vs. WT downregulated genes (*Figure 4b*, *Supplementary file 1-3*), in line with this well-established dystrophic abnormality (*Vallejo-Illarramendi et al., 2014*).

These results suggested alterations in proliferation, migration, and differentiation in both *Dmd*<sup>mdx</sup> and *Dmd*<sup>mdx-βgeo</sup> cells. To further assess these processes, we performed functional analyses of these three key myoblast functions in primary myoblasts isolated from gastrocnemii of dystrophic and control mice.

## Increased proliferative capacity of dystrophic myoblasts

The ability of myoblasts to proliferate is key to obtaining enough myogenic cells to repair muscle fibre damage. Given that the transcriptomic data indicated this mechanism as likely to be affected, we tested, in a BrDU incorporation assay, the dystrophic myoblasts' capacity to respond to the proliferative stimulus of the exposure to a sera-rich medium (*Figure 5a*). After six hours, 57% of *Dmd*<sup>mdx</sup> myoblasts have incorporated BrDU compared to 26% of WT cells (p=3.0e-3). 39.1% of *Dmd*<sup>mdx-βgeo</sup> cells were positive for BrDU compared to 24% of the corresponding C57Bl/6 controls (p=1.0e-2). This statistically significant two-fold increase in BrDU incorporation denotes an exaggerated response to a proliferative stimulus in both dystrophic cells compared to their respective WT controls. Although Dp71 expression was described as enhancing myoblast proliferation (*Farea et al., 2020*), we did not observe any negative effect on proliferation in dystrophin-null *Dmd*<sup>mdx-βgeo</sup> cells. On the other hand, this absence of differences between these two genotypes was consistent with the differential gene expression data described earlier.

## Significantly decreased chemotaxis of dystrophic myoblasts

In dystrophic myoblasts, several GO categories associated with migration were found to be overrepresented in the downregulated gene lists. We therefore assessed, using the Boyden chamber assay, the chemotaxis of dystrophic cells toward sera-rich medium and toward medium containing a cocktail of chemo-attractants (*Figure 5b*). *Dmd*<sup>mdx</sup> myoblasts chemotaxis towards both sera-rich and cytokine-containing medium was found to be significantly reduced, with respectively 28% (p=3.8e-2) and 21% (p=1.0e-2) cells penetrating through the membrane compared to WT. *Dmd*<sup>mdx-βgeo</sup> myoblasts also showed significantly reduced chemotaxis at 44% (p=5.0e-6) and 54 % (p=7.8e-4), compared to WT controls in serum-rich and cytokines media, respectively.

## Altered differentiation of *Dmd*<sup>mdx</sup> dystrophic myoblasts

Dystrophic myoblasts showed altered expression of several key genes involved in muscle differentiation (*Figure 1c*) and GO categories related to differentiation were enriched in downregulated gene lists (*Figure 4*, *Supplementary file 1-3*). Even though these analyses were performed in proliferating myoblasts, such alterations could indicate a distorted capacity for differentiation in dystrophic cells.

To assess this, differentiation of *Dmd*<sup>mdx</sup> myoblasts was compared against WT, using a three-dimensional (3D) culture, facilitating interactions closely resembling those occurring in muscles in situ.

Spheroids of dystrophic or WT cells (n=3 per group), were placed in a differentiation medium (t=0) and collected on days 0, 2, 4, and 6 to monitor their differentiation. We first established that after 6 days in such condition's spheroids differentiated sufficiently to express myosin heavy chain (*Figure 4—figure supplement 1*; supplementary movie https://youtu.be/1axNWeK-Yb0). Changes in differentiation were assessed by qPCR quantification of the expression of *Myh1*, *Myog*, and *Mymk*, gene markers of myoblast differentiation. Analysis of the time course of expression revealed a significantly altered pattern in *Dmd*<sup>mdx</sup> cultures. Specifically, *Mymk* (Time: DF = 3, *F*=7.520, p=3.4e-2.

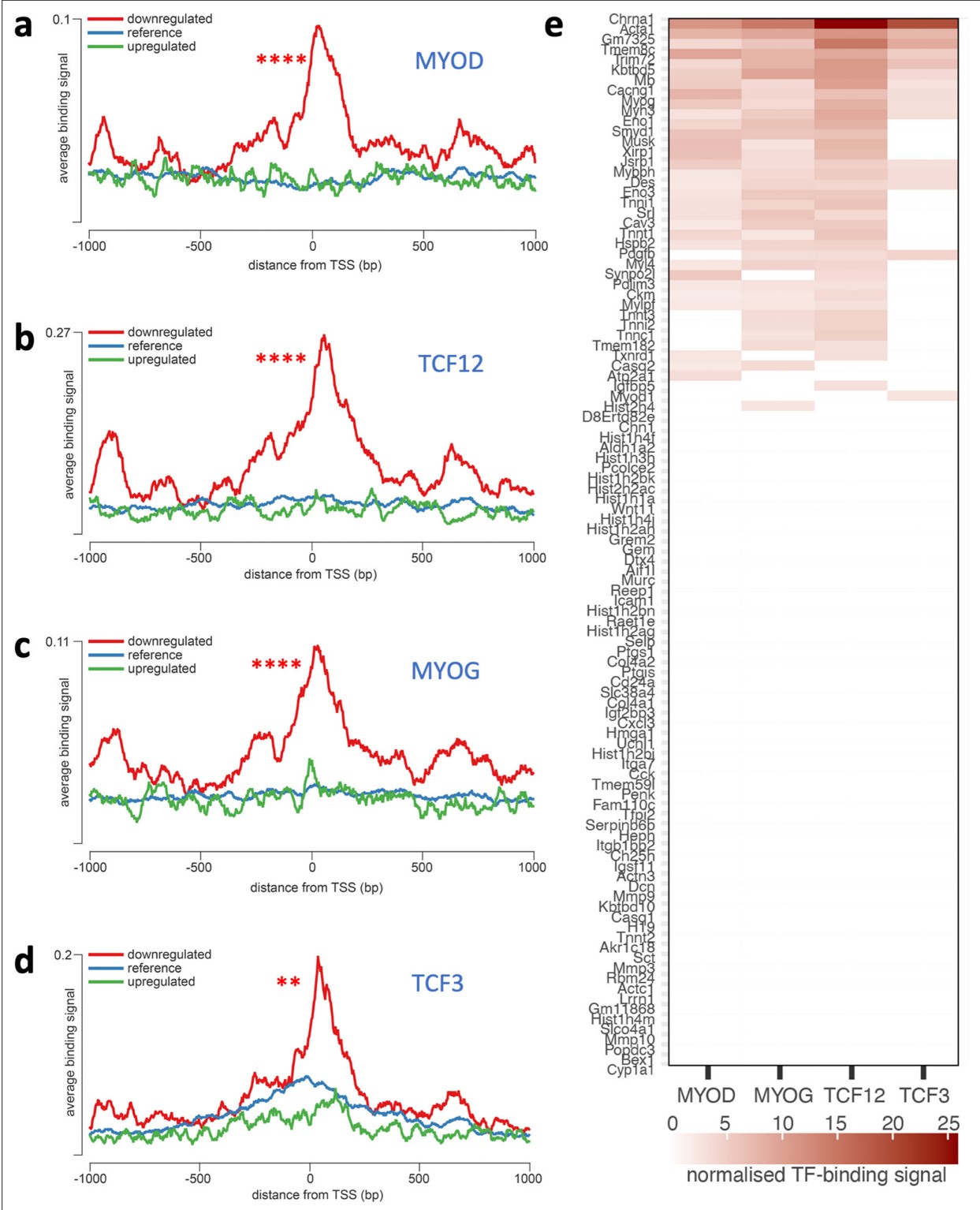

**Figure 3.** Overrepresentation of TF binding sites in genes downregulated in *Dmd*[mdx] myoblasts. Averaged ChIP-Seq signal histograms for each of the significantly overrepresented TFBS in downregulated genes. Histograms are centred around the transcription start site (TSS) of each gene. x-axis – distance upstream and downstream from the TSS of each gene, y-axis - ChIP-Seq signal for each location averaged for all gene promoters in the submitted list; red line - downregulated genes, blue line - upregulated genes, green line - reference signal (based on 1000 random gene promoters); Bonferroni-corrected p-value of TF signal overrepresentation vs reference: ***<1.0e-3; **<1.0e-2. (**a**) MYOD (adjusted p-value: 2.9e[−21], track GEO accession GSM915165), (**b**) TCF12 (1e[−19], GSM915178), (**c**) MYOG (5.6e[−19], GSM915164); (**d**) TCF3 (0.0033, GSM915177); (**e**) Heatmap of normalised TF-

*Figure 3 continued on next page*

*Figure 3 continued*

binding signal for each of the TFs overrepresented in genes downregulated in *Dmd*^mdx^ myoblasts. Genes that did not show significant binding above background (as provided by the seqinspector tool) were assigned TF binding signal of 0.

Group: DF = 1, $F$=50.40, p=2.1e-3), *Myog* (Time: DF = 3, $F$=9.811, p=1.7e-2. Group: DF = 1, $F$=11.72, p=2.7e-2) and *Myh1* (Time: DF = 3, $F$=24.97, p=8.0e-4. Group: DF = 1, $F$=66.95, p=1.2e-3) were found to have their expression statistically significantly increased (two-way ANOVA) in dystrophic spheroids, when compared to WT (*Figure 5c*).

Also, in our hands, *Dmd*^mdx^ myogenic cells in 2D cultures formed myotubes within ≈ 5 days, while WT myoblasts required minimum 7 days in the differentiation medium (unpublished), which is in agreement with reports that dystrophic myoblasts differentiate faster (*Yablonka-Reuveni and Anderson, 2006*).

It is worth noting that these increases in *Myh1*, *Myog,* and *Mymk* expression in differentiating cells contrast with their expression profiles in proliferating myoblast, where dystrophic cells showed consistently lower expression levels of these markers (*Figure 1—figure supplement 1*; *Supplementary file 1-1*).

## Dystrophic myoblast cell line reproduces transcriptome anomalies found in primary cells

Although multiple replicates of primary myoblasts from the same genotype had similar transcriptome profiles (*Figure 1*), and primary cultures were free from other cell types, myoblasts isolated from *Dmd*^mdx^ muscle have been exposed to the dystrophic niche and some of the alterations might result from environmental factors. Therefore, we investigated whether alterations found in primary cells also occur in an established myoblast cell line. We used the SC5 (*Dmd*^mdx^) and IMO (WT) cells, both derived from the H2Kb-tsA58 mice (*Morgan et al., 1994*) and thus having an identical genetic background. Comparison of RNA-Seq data showed strong segregation of samples according to genotypes (*Figure 6a*) and a clear dysregulation in gene expression (*Figure 5b*, *Supplementary file 1–4*). The SC5 dystrophic cell line transcriptome had significantly more dysregulated genes compared to IMO controls (*Supplementary files 1–4*) than the primary *Dmd*^mdx^ myoblasts compared to WT cells (*Supplementary file 1-1*). Importantly, despite this transcriptome drift expected in cells maintained long-term in culture (*Kim et al., 2018*), the key alterations seen in the transcriptome of dystrophic primary cells were also found in the established dystrophic cell line. *Myod1, Myog, Mymk, Des* and 69 other genes (*Figure 6*, *Supplementary file 1–4*) were found significantly dysregulated in dystrophic cells, and an overlap in GO (*Figure 6c and d*; *Supplementary file 1*) demonstrated that the impact of the mdx mutation is present, with 61 GO categories for biological processes being enriched in the downregulated genes lists in both primary and cell line myoblasts carrying the *Dmd*^mdx^ allele.

Further analysis of these 61 GO categories (*Supplementary file 1-5*) revealed that all those altered are among the most significantly dysregulated both in terms of adjusted p-values and fold enrichment in both primary and established cells, involving very relevant processes and functions, such as 'striated muscle tissue development' and 'striated muscle cell differentiation' (*Figure 6*, *Supplementary file 1-5*).

Moreover, analysis of overrepresented TF binding sites on promoter regions of the dysregulated genes found transcripts downregulated in the SC5 cells to have significantly higher than background signal for MyoD, MYOG, and TCF12 (*Supplementary file 2G*), in clear agreement with TFBS overrepresented in primary *Dmd*^mdx^ myoblasts. Likewise, genes upregulated in SC5 did not exhibit statistically significant overrepresentation of the binding signal of any TF (*Supplementary file 2F*).

Finally, in line with these transcriptomic similarities, the dystrophic myoblast cell line exhibited the impaired chemotaxis phenotype, identical to that identified in primary dystrophic myoblasts (*Figure 5—figure supplement 1*).

## Human and mouse dystrophic primary myoblasts exhibit corresponding transcriptome changes

Comparison of differential gene expression data from human DMD and healthy primary myoblasts also showed significant transcriptomic alterations with clear segregation between samples according

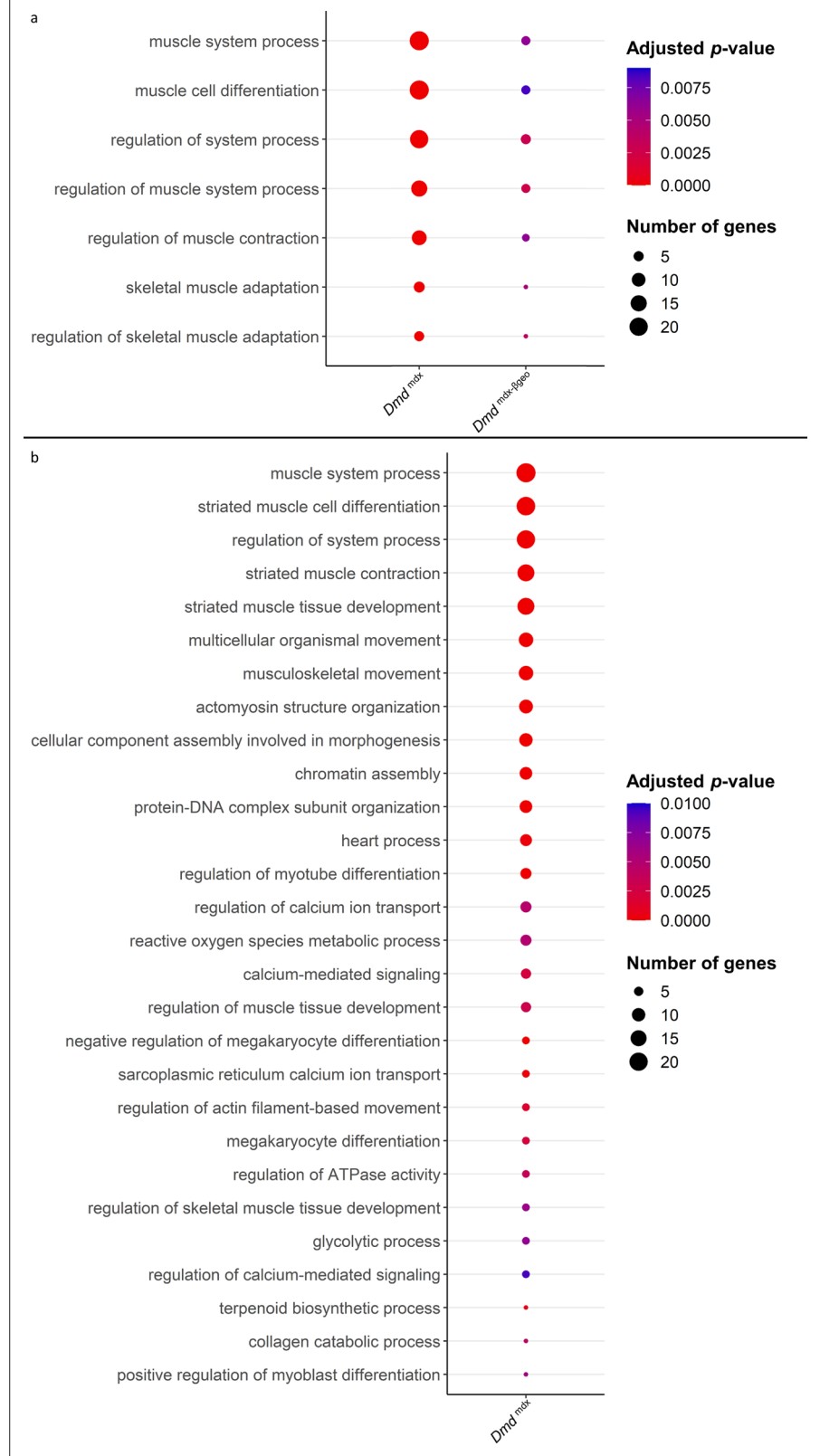

**Figure 4.** Results of GO category enrichment analysis for biological processes on significantly downregulated gene lists from *Dmd*mdx and *Dmd*mdx-βgeo versus WT differential gene expression analyses. (**a**) Bubble plot of the overlapping GO categories enriched in *Dmd*mdx and *Dmd*mdx-βgeo versus WT downregulated genes lists. Bubble size represents the number of genes from the downregulated gene lists belonging to the GO category and the bubble

*Figure 4 continued on next page*

*Figure 4 continued*

colour depicts the adjusted *p*-value from red (0) to blue (8.0e-3). (**b**) Bubble plot of GO categories enriched in *Dmd*^mdx versus WT downregulated genes lists following REVIGO redundancy filtering (see Methods). Bubble size represents the number of genes from the downregulated gene lists belonging to the GO category and the bubble colour depicts the adjusted p-value from red (0) to blue (1.0e-2).

The online version of this article includes the following figure supplement(s) for figure 4:

**Figure supplement 1.** Immunodetection of the sarcomeric myosin heavy chain 4 in *Dmd*^mdx and BL/10 in differentiated spheroids.

---

to genotypes (*Figure 7a*) and a greater number of significantly downregulated genes (334) compared to upregulated ones (86) (*Figure 7b*, *Supplementary file 1-6*). This profile was very similar to the primary mouse myoblast data (*Figure 1*, *Supplementary file 1-1*).

Comparison of log2 fold changes of significantly dysregulated genes in human DMD myoblasts and their orthologous counterparts in mouse primary *Dmd*^mdx myoblasts revealed that genes significantly downregulated in one species tend to also be downregulated in the other, as shown by more genes in the lower-left quadrant of *Figure 7c*, compared to other quadrants (Fisher's exact test: p=2.0e-4). With *r* values ranging from 0.2357 to 0.4688, depending on which gene populations are compared (*Figure 7c and d*), the magnitude of gene expression changes between species was equivocal.

However, a comparison of the human GO enrichment for biological processes in the significantly downregulated gene list to its mouse counterpart returned a strikingly similar result, with 61 overlapping categories (*Figure 7d and e- Supplementary file 1-7*). Of these, 49 also overlap with the categories in the mouse dystrophic myoblasts cell line. Again, the most significantly altered categories (in terms of adjusted p-value) that overlap in both species were associated with muscle cell development and differentiation (*Figure 7c*, *Supplementary file 1-7*).

When comparing the list of significantly downregulated genes in *Dmd*^mdx versus WT (*Supplementary file 1-1*) and human DMD versus healthy human myoblasts (*Supplementary file 1-7*), taking into account only one-to-one orthologs, 41 genes are present in both lists including key myogenic program actors such as *Myog*, *Mymk*, *Myod1*, *Des*, *Smyd1*, *Acta1*, *Actc1*, and *Atp2A1*.

Interestingly, other genes of interest found to be dysregulated in primary mouse myoblasts were found to have non-orthologous counterparts dysregulated in human primary DMD myoblasts: *COL11A1*, *COL14A1* and *COL15A1* for *Col2a1*, *Col4a1*, *Col4a2*, and *Col8a2* or *MMP2* for *Mmp3*, *Mmp9*, and *Mmp10*.

Finally, a three-way comparison of the GO enrichment for biological processes in the downregulated genes for dystrophic primary human myoblasts, primary mouse myoblasts and the dystrophic mouse myoblast cell line showed an important overlap: 49 categories were found enriched in all three datasets, with further 12 enriched in human and mouse primary cells, 12 in mouse primary and mouse cell line datasets and finally 47 categories enriched in both primary human and mouse cell line dystrophic myoblasts (*Figure 8a*). In the 49 categories enriched in all three datasets, many are relevant to the disease and consistent with the cell function anomalies described earlier. Specifically, categories such as 'muscle cell differentiation', 'muscle system process', 'regulation of myoblast differentiation' and 'regulation of muscle system process' are very significantly enriched in the downregulated gene lists for all datasets (*Figure 8b*).

Alterations in transcriptomes of human and mouse dystrophic myoblasts are strikingly analogous, with downregulation of gene expression predominating in both, reduction in key markers of the muscle program and relevant common GO categories being enriched. Importantly, our data provide a molecular underpinning for previously reported abnormalities in proliferation, migration and differentiation of human DMD myoblasts (*Witkowski and Dubowitz, 1985*; *Melone et al., 2000*; *Nesmith et al., 2016*; *Sun et al., 2020*).

## Metabolic pathway alteration in DMD primary myoblasts

Skeletal muscle metabolism is a significant variable (*Zurlo et al., 1990*), as metabolic dysfunction can impair skeletal muscle regeneration, and DMD affects cellular energetics in muscle progenitors (*Matre et al., 2019*). Therefore, we investigated the metabolic alterations in human DMD myoblasts. To this end, we built a dystrophic genome-scale metabolic model, following our previous pipelines for generating context-specific metabolic models using omics data (*Magazzù et al., 2021*). *Figure 9* and

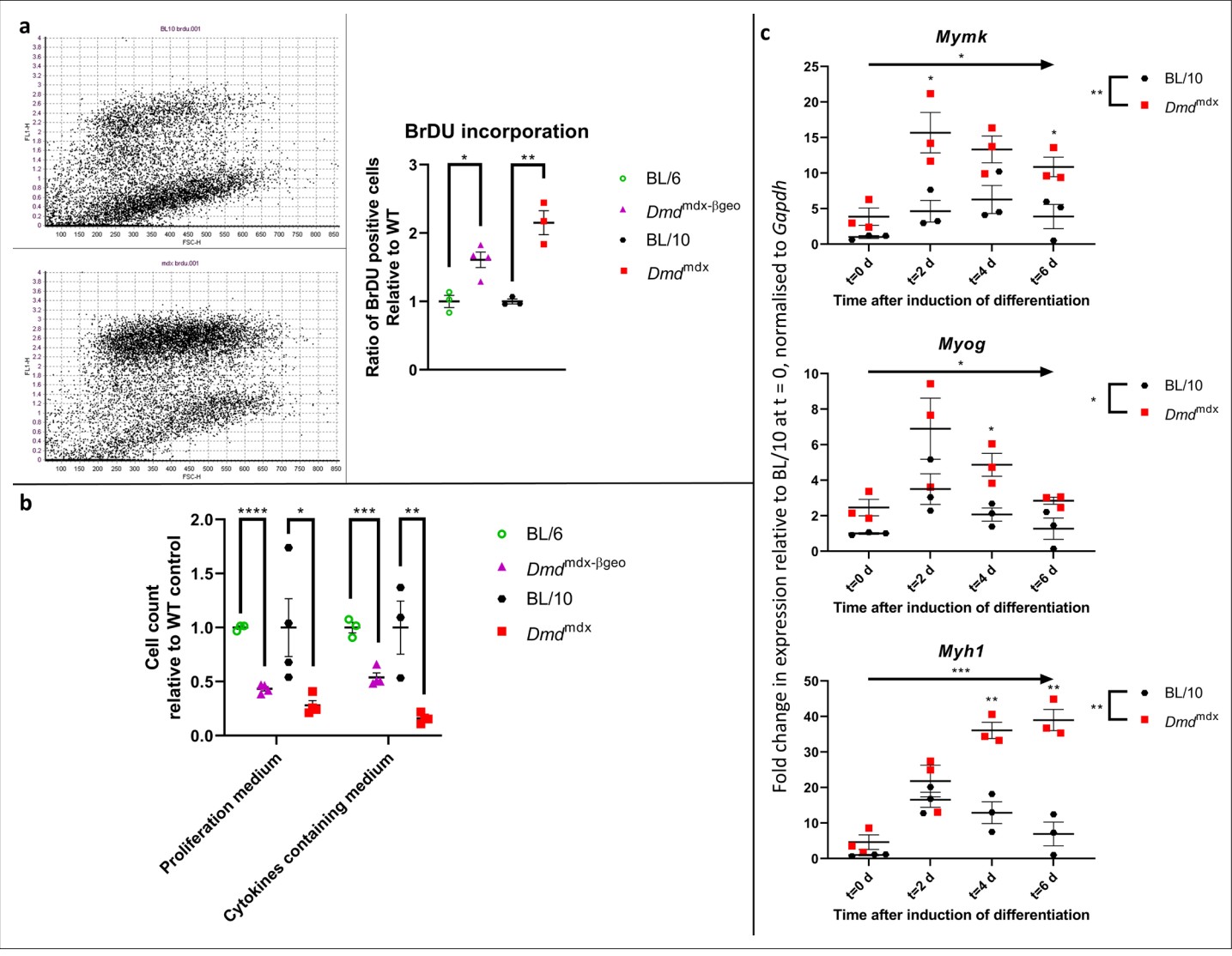

**Figure 5.** Dystrophic primary myoblasts show functional abnormalities. (**a**) Proliferation analysis in myoblasts from $Dmd^{mdx}$, $Dmd^{mdx-\beta geo}$, C57BL/10 and C57BL/6 mice using a BrdU incorporation assay. (Left) Representative examples of BrdU flow cytometric dot plots from C57BL/10 and $Dmd^{mdx}$ cells after 6 hr of incorporation, with FL-1 channel corresponding to BrdU fluorescence intensity and FSC channel denoting the cell size. (Right) Graph showing significantly increased proliferation of dystrophic myoblasts presented as ratio of BrdU-positive dystrophic cells relative to the respective wild type controls. Errors bars represent mean ± SEM, n=3 or 4, *=p ≤ 5.0e-2, **=p < 1.0e-2 (Student's unpaired t-test). (**b**) Cell chemotaxis analysis in myoblasts from $Dmd^{mdx}$, $Dmd^{mdx-\beta geo}$, and respective wild-type controls. Cells were seeded on a trans-well insert and allowed to penetrate towards the bottom well containing either proliferation medium or serum-free medium complemented with cytokines. Graph shows significantly reduced chemotaxis of dystrophic myoblasts represented as the relative number of cells present on the well-side of the membrane after 12 hr. Error bars represent mean ± SEM, n=3 or 4, *=p < 5.0e-2, **=p < 1.0e-2 and ***=p < 1.0e-3 (Student's unpaired t-test). (**c**) Altered expression of differentiation markers in $Dmd^{mdx}$ compared to C57BL/10 myoblasts. Results of qPCR expression analyses of $Myh1$, $Myog$, and $Mymk$ markers at specific timepoints over the 6-day period of spheroid differentiation. Individual values for biological replicates normalised to $Gapdh$ expression and relative to wild type values at the commencement of differentiation (t=0) are shown. Error bars represent mean ± SEM, n=3, *=p ≤ 5.0e-2, **=p < 1.0e-2 and ****=p < 1.0e-4 (two-way ANOVA was used to determine the statistical significance between groups and time-points over the 6-day period and Fisher's LSD was used to determine which timepoints exhibited a significant difference between groups).

The online version of this article includes the following figure supplement(s) for figure 5:

**Figure supplement 1.** Decreased cell chemotaxis of dystrophic established mouse myoblasts.

*Figure 9—figure supplement 1* illustrate the metabolic activity fold change (expressed in log2FC) across reactions in the enzyme-constrained human metabolic model. We differentiated the metabolic pathways into upregulated and downregulated groups (**Supplementary file 3**). We found a significant change in the rate of glycolysis/gluconeogenesis (log2FC = 4.8), leukotriene metabolism (log2FC =

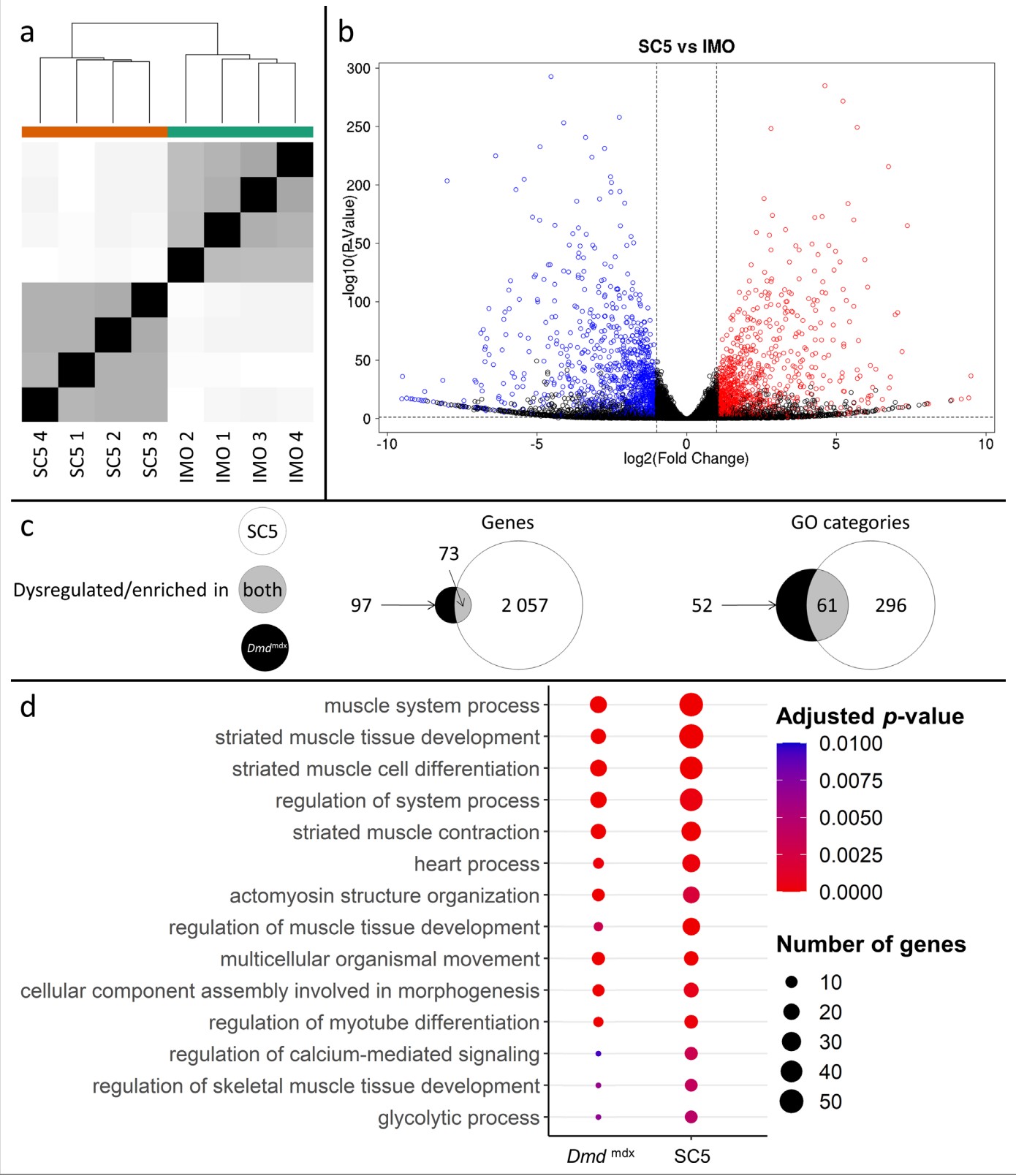

**Figure 6.** Correlation of significantly dysregulated gene expressions between mouse dystrophic and WT myoblast cell lines (SC5 vs.IMO) and of significantly enriched GO categories in the downregulated gene list for primary *Dmd*mdx myoblasts and corresponding cell lines. (**a**) Sample-based hierarchical clustering heatmap, colour represents dissimilarity between samples based on the Euclidean distance, from black for identical samples to white for more distinct samples. (**b**) Volcano plot, showing the results of differential expression analysis between SC5 and IMO. Circles represent

*Figure 6 continued on next page*

*Figure 6 continued*

individual genes, with their colour representing significance and direction of dysregulation (blue-significantly downregulated; red-significantly upregulated; black-not significantly dysregulated). Circle position represents fold change on a log2 scale for the x-axis and adjusted *p*-value on a -log10 scale for the y-axis. Horizontal dotted line represents the threshold of an adjusted *p*-value of 5.0e-2 while the vertical doted lines represent the threshold for the absolute log2 fold change of 1. (**c**) Venn diagrams representing the overlap between genes dysregulated in SC5 and primary *Dmd*^mdx myoblasts vs. WT as well as the overlap between GO categories for biological processes overrepresented in both downregulated gene lists. Number in the white circle enumerates dysregulated genes or enriched GO categories in SC5 vs. IMO, black circle those of *Dmd*^mdx vs. WT and grey area numbers represent overlap between these two sets. (**d**) Bubble plot of the overlapping GO categories following redundancy filtering applied to the 61 categories identified (panel c). Bubble size represents the number of genes from the downregulated gene lists belonging to the category and bubble colour represents the adjusted p-value from red (0) to blue (1.0e-2).

4.754), propanoate metabolism (log2FC = 5.657), mitochondrial beta-oxidation of branched-chain fatty acids, odd-chain fatty acids, and di-unsaturated fatty acids (n-6) (log2FC = –1.187, log2FC = –0.8295 and log2FC = –0.655) (*Figure 10a*).

Similarly, the activity of phosphoglycerate mutase 2 and 2-phospho-D-glycerate proteins is significantly increased in the glycolysis/gluconeogenesis metabolic pathway (*Figure 10*, *Supplementary file 3*). The glycolytic enzyme, phosphoglycerate, catalyses the interconversion of 3-phosphoglycerate and 2-phosphoglycerate in the presence of 2,3-bisphosphoglycerate (*Durany et al., 2002*). The glycosphingolipid biosynthesis-lacto and neolacto series pathways are also downregulated in DMD (log2FC = –0.49), decreasing the production of type I B glycolipid, sialyl-3-paragloboside metabolites.

Mitochondrial propanoate metabolism is observed to have upregulated activity in DMD cells (log2FC = 5.657) (*Figure 9*) indicating an increase in the activity of propionyl-CoA synthetase in mitochondria, which can impair myogenic differentiation (*Lagerwaard et al., 2021*).

DMD is characterised by the decrease of mitochondrial aconitase (ACO2) *Capitanio et al., 2020*, and the downregulation of isocitrate hydro-lyase citrate cycle metabolic reactions (r0426) in the TCA cycle indicates a significant decrease of the aconitase metabolite in DMD myoblasts. This suggests that the interconversion of citrate to isocitrate as part of the citric acid cycle is carried out at a reduced rate, therefore directly impairing carbohydrate and energy metabolism. We also found that the upregulation of glycine-N-acetyltransferase reaction in Phenylalanine metabolism increases the production of phenylacetyl-CoA and glycine in DMD myoblasts. The upregulation of phenylalanine in skeletal muscle is associated with decreased mTOR activity, a key regulator of cell growth and anabolism (*Vendelbo et al., 2014*).

Alterations in the activity of other mitochondrial pathways and reactions were also found (*Figure 9a and b* respectively). Further reaction-level alterations in pyruvate metabolism, glycolysis/gluconeogenesis, transport reactions, glycosphingolipid biosynthesis-lacto and neolacto series metabolic pathways are shown in *Figure 9* and *Figure 9—figure supplement 1*.

Thus, these molecular and fucntional myoblast abnormalities identified across mouse and human dystrophic cells and in cell lines are the consequence of DMD gene mutations.

## Discussion

DMD presents in muscle stem cells, where the loss of dystrophin affects asymmetrical cell divisions (*Dumont et al., 2015b*). Subsequently, the absence of dystrophin during myotube differentiation causes typical dystrophic abnormalities such as altered calcium homeostasis and creatine kinase leakage (*Shoji et al., 2015*). We hypothesised that myoblasts originating from dystrophic stem cells, which give rise to myotubes, are also affected. Indeed, we demonstrate here for the first time that, in proliferating human and mouse myoblasts, the absence of *DMD* gene expression results in major abnormalities. Our data combining global RNA-Seq and functional analyses demonstrate that DMD directly affects myoblasts.

The key alterations concerned proliferation, chemotaxis, and differentiation. Importantly, synchronisation of these three processes, involving exit from the cell cycle, migration to the site of damage, activation of the skeletal muscle-specific genes and cell fusion, is essential for muscle development and regeneration. Ultimately, myoblasts are the effector cells of muscle growth and repair, and it is the failure of muscle regeneration that drives DMD progression.

These defects are cell autonomous rather than caused by the inflammatory environment of the dystrophic niche because they persist in the dystrophic cell line maintained long-term in culture:

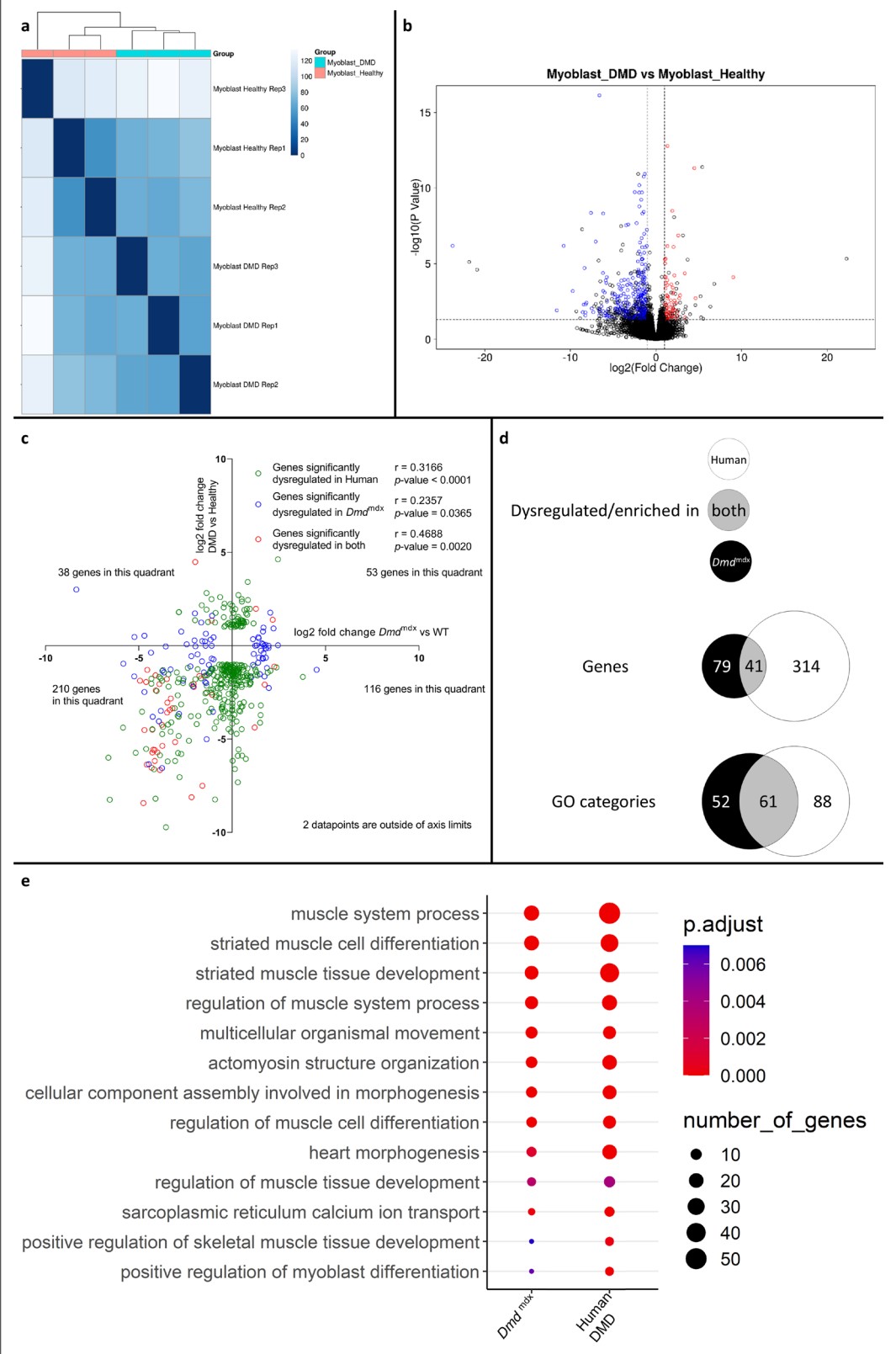

**Figure 7.** Correlation of significantly dysregulated gene expressions between human and mouse dystrophic and WT primary myoblasts and of the corresponding significantly enriched GO categories in the downregulated gene list for the human and Dmd[mdx] myoblast data. (**a**) Sample-based hierarchical clustering heatmap, represented as described before. (**b**) Volcano plot, showing the results of differential expression analysis between DMD and

*Figure 7 continued on next page*

*Figure 7 continued*

healthy myoblasts. Circles represent individual genes, with their color representing significance and direction of dysregulation (blue-significantly downregulated; red-significantly upregulated; black-not significantly dysregulated). Circle position represents fold change on a log2 scale for the x-axis and adjusted p-value on a -log10 scale for the y-axis. Horizontal dotted line represents the threshold of an adjusted p-value of 5.0e-2 while the vertical doted lines represent the threshold for the absolute log2 fold change of 1. (**c**) log2 fold change values (*Dmd*mdx versus WT on the x-axis and human DMD versus WT on the y-axis) for genes significantly dysregulated in one or both dystrophic vs. WT primary myoblasts. Pearson's correlation coefficients (*r*) and p-values are shown. Green circles represent genes significantly dysregulated in DMD, blue circles represent genes significantly dysregulated in *Dmd*mdx vs. WT and red circles represent genes significantly dysregulated in both. The number of genes found in each quadrant is shown. (**d**) Venn diagrams representing the overlap between genes dysregulated in primary human DMD and *Dmd*mdx myoblasts vs. WT as well as the overlap between GO categories for biological processes overrepresented in both downregulated gene lists. Number in the white circle enumerates dysregulated genes or enriched GO categories in human DMD vs. healthy, black circle those of *Dmd*mdx vs. WT and grey area numbers represent overlaps between these two sets. (**e**) Bubble plot of the overlapping GO categories following redundancy filtering applied to the 61 categories identified (panel d). Bubble size represents the number of genes from the downregulated gene lists belonging to the GO category and the bubble colour depicts the adjusted p-value from red (0) to blue (7.0e-3).

established cells reproduced the key transcriptomic anomalies and were described before to have functional alterations found in primary cells (*Róg et al., 2019*).

Moreover, alterations are shared between mouse and human DMD myoblasts, despite the significant heterogeneity within human samples (*Mournetas et al., 2021*; *Choi et al., 2016*). Therefore, these dystrophic anomalies manifest irrespective of genetic backgrounds and across species, which confirms their significance. Furthermore, altered proliferation, migration and differentiation in DMD myoblasts have been reported (*Witkowski and Dubowitz, 1985*; *Melone et al., 2000*; *Nesmith et al., 2016*; *Sun et al., 2020*) and our data provide the molecular underpinning for these abnormalities.

DMD pathology being intrinsic to the myoblast has been proposed previously (*Blau et al., 1983*) and there has been growing evidence that *DMD* mutations cause a range of cell-autonomous abnormalities in human and mouse myogenic cells and myoblasts, which affect cell division, differentiation, energy metabolism and signalling (*Yablonka-Reuveni and Anderson, 2006*; *Yeung et al., 2006*; *Sacco et al., 2010*; *Sacco et al., 2010*; *Young et al., 2020*; *Young et al., 2020*; *Dumont et al., 2015b*; *Onopiuk et al., 2009*; *Onopiuk et al., 2015*; *Dumont et al., 2015a*; *Young et al., 2015*). Interestingly, *Dmd*mdx cell migration found reduced under physiological stimuli (this work) can be increased in the inflammatory environment (*Young et al., 2020*). In fact, altered proliferation and migration resulting in highly metastatic phenotypes, have been associated with the loss of Dp427 in tumours featuring myogenic differentiation (*Wang et al., 2014*).

Our team recently demonstrated that *DMD* mutations evoke marked transcriptome and miRNome dysregulations early in human muscle cell development (*Mournetas et al., 2021*). In that study, expression of key coordinators of muscle differentiation was dysregulated in proliferating dystrophic myoblasts, the differentiation of which was subsequently also found to be altered, in line with the mouse cells studied here (*Figure 5c*).

A significant number of downregulated transcripts in dystrophic myoblasts are controlled by the same TFs, most notably MyoD and MYOG. Given that MYOG itself is regulated by MyoD, the significant downregulation of *Myod1* expression found here appears to be the key trigger mechanism of the dystrophic abnormality in myoblasts: the lower levels of *Myod1* transcript strongly correlate with downregulated expression of genes controlled by this TF, and which are key coordinators of skeletal muscle repair. Consequently, altered expression of myogenic regulatory factors in dystrophic myoblasts could have multiple consequences. Indeed, complete *Myod1* ablation in *Dmd*mdx has been shown to exacerbate the disease due to a diminished capability for muscle regeneration (*Megeney et al., 1999*; *Schuierer et al., 2005*). Downregulation of MyoD, which inhibits the cell cycle, could explain the increased proliferation in dystrophic cells. The *Myod1-Mymk-Mymx* (Myomaker/ Myomixer) axis altered here was found to determine myotube formation (*Zhang et al., 2020*). Given that *Pax3* inhibits myogenic differentiation of myoblasts (*Epstein et al., 1995*), its downregulation in dystrophic cells could also contribute to this accelerated differentiation pattern observed here.

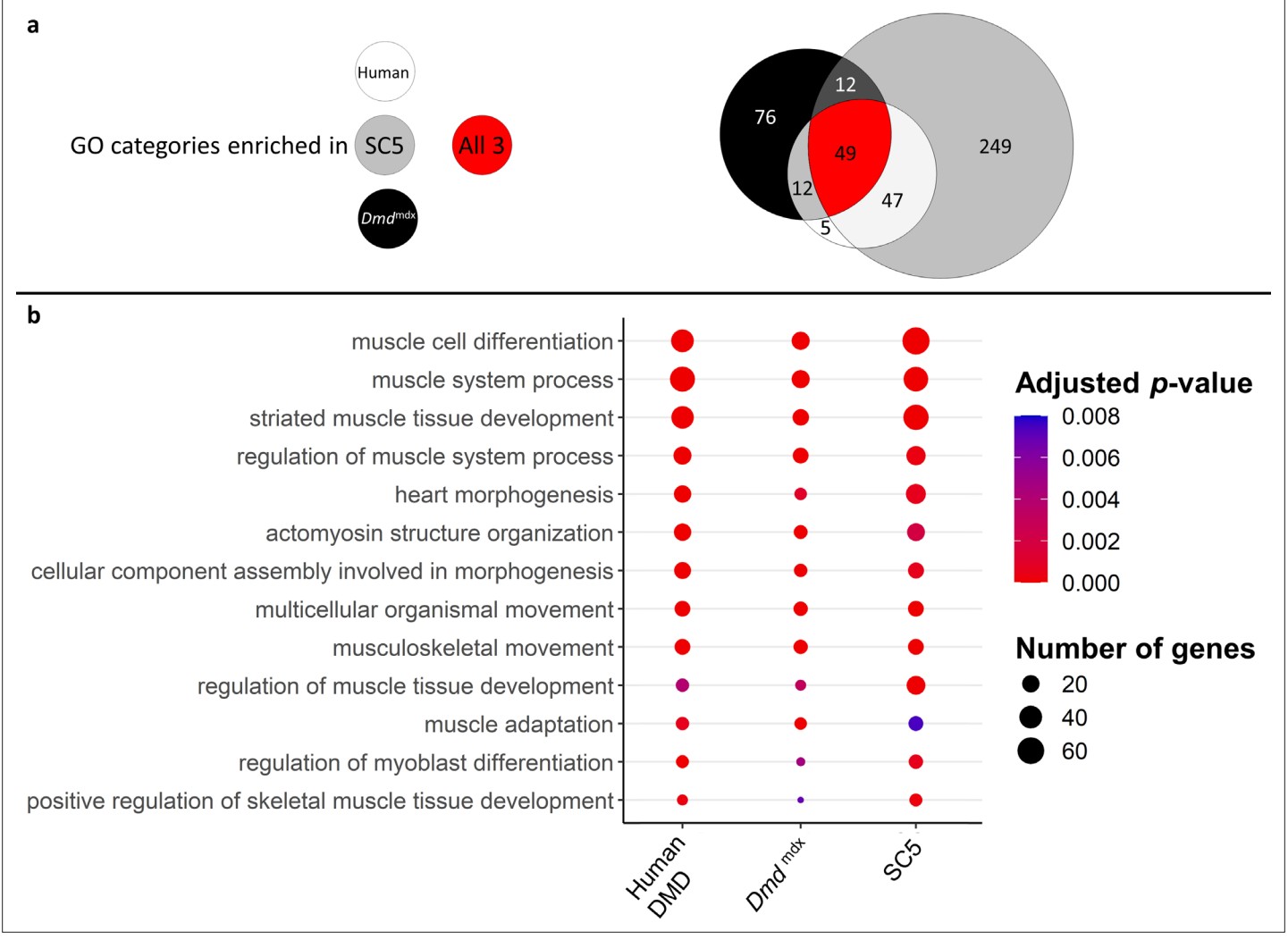

**Figure 8.** Correlation of enriched GO categories between primary *Dmd*^mdx mouse myoblasts, *Dmd*^mdx mouse myoblast cell line, and human primary DMD myoblasts. (**a**) Venn diagram representing the overlap between GO categories for biological processes overrepresented in downregulated gene lists in primary human DMD, *Dmd*^mdx myoblasts and established *Dmd*^mdx myoblast cell line (SC5). Numbers in each area enumerate enriched GO categories in individual datasets and those overlapping (see the graphic legend). (**b**) Bubble plot of the overlapping GO categories following redundancy filtering applied to the 49 categories identified (panel a). Bubble size represents the number of genes from the downregulated gene lists belonging to the given GO category and bubble colour depicts the adjusted *p*-value from red (0) to blue (8.0e-3).

GO categories concerning calcium homeostasis and signalling, known to be altered in dystrophic muscles across species, were also found altered in dystrophic myoblasts: calsequestrins 1 and 2 levels being significantly downregulated, in line with previous findings in the dystrophic muscle *Waugh et al., 2014*; *Doran et al., 2004* and cell lines (*Róg et al., 2019*; *Onopiuk et al., 2015*).

Given that Dp71 dystrophin is found in undifferentiated myogenic cells (*Howard et al., 1999*), and that in development shorter dystrophins have been associated with proliferation and migration, and long isoforms with terminal commitment (*Hildyard et al., 2020*), we hypothesized that eliminating expression of Dp71 may exacerbate dysfunctions in myogenic cells. Surprisingly, we found that both *Dmd*^mdx and *Dmd*^mdx-βgeo cells had similar abnormalities and, contrary to Dp71 overexpression increasing myoblast proliferation (*Farea et al., 2020*), BrdU incorporation in dystrophin-null myoblasts was still significantly increased compared to WT. Interestingly, *Dmd*^mdx myoblasts, having a point mutation in exon 23, expressed significantly lower levels of Dp71 (*Appendix 1—figure 1*), whose expression is driven by a promoted located over 40 exons downstream of the mutation. But whether this downregulation is sufficient to mimic its total loss is unknown.

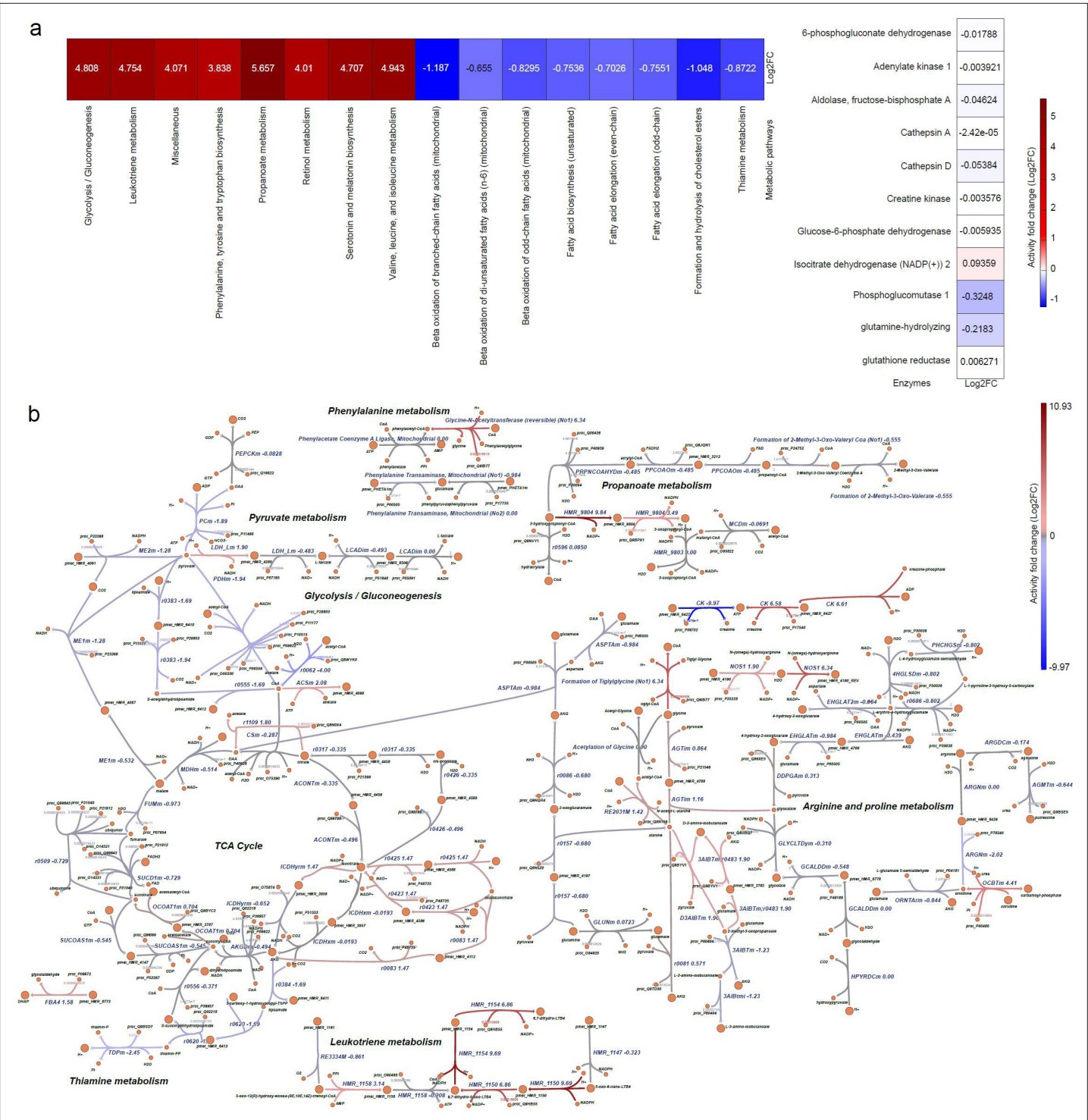

**Figure 9.** *Metabolic pathway alterations in DMD primary myoblasts.* The differential log2FC of DMD-specific metabolic model was compared with respect to the healthy Human metabolic model. (**a**) Heatmap representing the differential metabolic activity of the pathways and proteins. (Left) Differential flux change of metabolic pathways. (Right) Flux fold change of the activity of enzymes, represented by proteins. (**b**) Visualisation of the alteration in the reactions in metabolic pathways. The red colour represents the upregulated or overactive metabolic pathways, and the blue colour represents the downregulated or underactive metabolic pathways. The full names of the reactions presented in *Figure 10* are provided in *Supplementary file 4*.

The online version of this article includes the following figure supplement(s) for figure 9:

**Figure supplement 1.** Differential metabolic pathway and reaction analysis.

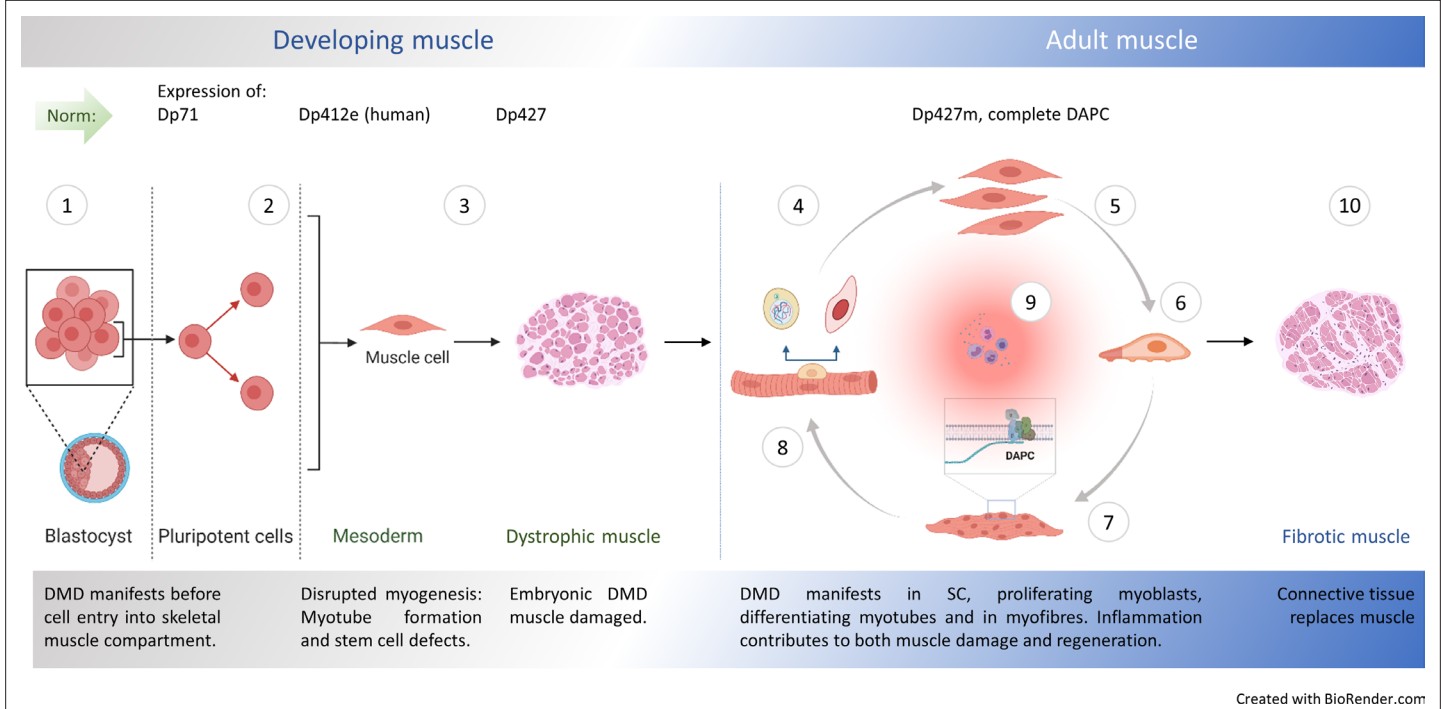

**Figure 10.** Natural history of DMD. ① Embryonic stem cells (ES) express dystrophin Dp71 *Rapaport et al., 1992* but its loss is not linked with arrested development. Yet, minor admixture of ES lacking Dp427 results in severe dystrophy *Gonzalez et al., 2017*: Defects in myogenic stem cells (SC) descendant from DMD ES must be responsible, because in myofibre syncytium, lack of dystrophin caused by small proportion of DMD nuclei would be compensated and damage prevented. Early mesoderm committed human cells express Dp412e *Massouridès et al., 2015* and ② DMD manifests before entry into the skeletal muscle compartment *Mournetas et al., 2021*. ③ DMD abnormalities are found in dystrophic human *Toop and Emery, 1977*; *Emery, 1977*; *Vassilopoulos and Emery, 1977*, dog *Nguyen et al., 2002*, zebrafish *Bassett et al., 1977* and mouse embryos, with stem cell dysfunctions (hyperproliferation and death), disrupted myotube formation *Merrick et al., 2009* and fibrosis as intrinsic features in developing muscle *Mournetas et al., 2021*. Typical dystrophic abnormalities (e.g. high serum CK) occur in new-borns years before diagnosis *van Dommelen et al., 2020*; *Pescatori et al., 2007*. In adult muscle, healthy myogenic cells, myotubes and myofibres express Dp427 and DAPC assembles in differentiating fibres, supporting their functions. Impaired muscle regeneration occurs due to: ④ Cell-autonomous defects in dystrophic satellite cells (SC) affecting asymmetric cell divisions *Sacco et al., 2010*; *Dumont et al., 2015b*; *Dumont et al., 2015a*, ⑤ hyperproliferation of dystrophic myoblasts (this work) and purinergic hypersensitivity *Górecki, 2016*; *Young et al., 2012*, ⑥ anomalous migration of DMD myoblasts: reduced under physiological conditions (this work) but augmented in the inflammatory environment *Young et al., 2018* and ⑦ accelerated dystrophic myoblast differentiation (this work and Yablonka-Reuveni and Anderson, 2006) with defects in differentiating myotubes lacking Dp427 *Shoji et al., 2015*. ⑧ Resulting mature muscle lacking DAPC show contraction-induced injury but dystrophin ablation in fully differentiated myofibres does not trigger muscular dystrophy *Rader et al., 2016*; *Ghahramani Seno et al., 2008*. ⑨ Chronic inflammation exacerbates muscle damage *Sinadinos et al., 2015*; *Howard et al., 2021*; *Al-Khalidi et al., 2018*, but specific inflammatory mediators are also needed for regeneration *Tidball, 2017*. ⑩ Muscle fibrosis intensifies, as the progressing loss of regenerative potential results in myofibres being replaced by the connective tissue.

Combined, these data bring to light the significance of full-length dystrophin defects in undifferentiated muscle cells, the consequences of which correspond to the impaired muscle regeneration occurring in this disease.

Clearly, the important question is the mechanism altering the myoblast phenotype. Dp427 in satellite cells appears to control asymmetrical cell division (*Dumont et al., 2015b*), which might suggest an interaction problem also in myoblasts.

However, even healthy myoblasts do not express Dp427 protein at levels detectable by Western blotting: an observation that contributed to the belief that myoblasts are not affected by *DMD* mutations. Although our attempts at immunolocalization in unsynchronised primary myoblasts failed to detect Dp427 (*Appendix 1—figure 2*), a precise spatio-temporal requirement for small amounts of full-length dystrophin, analogous to its role in satellite cells *Dumont et al., 2015b* or at neuronal synapses *Lidov et al., 1990* is a possibility that should be studied. Importantly, WT myoblasts express the full-length transcript, which is depleted in dystrophic cells. It is worth noting that this 14 kb mRNA, which transcription time may be as long as 16 hr *Tennyson et al.,*

*1995*, is produced in rapidly dividing myoblasts. Its expression is therefore unlikely to be meaningless. Indeed, downregulation of full-length dystrophin occurs across various malignancies and is associated with alterations of molecular pathways that were identical with alterations found in Duchenne muscles. Moreover, low DMD gene expression in tumours was associated with poor patient survival (https://www.biorxiv.org/content/10.1101/2022.04.04.486990v3). Clealry, dystrophin mis-expression is far more consequenctial and can affect many more cells and tissues than previously understood.

Is it possible that loss of the full-length dystrophin expression may trigger a mechanism not involving its protein product? Considering the increasing number of human pathologies caused by RNA, including another dystrophy – *dystrophia myotonica Kino et al., 2018*, it is not inconceivable that loss of a large 14 kb transcript made up of 79 exons may trigger a novel RNA-mediated disease process *Chan et al., 2018*. Abnormalities in *Dmd*<sup>mdx</sup> cells, that harbour just a point mutation, indicate that the patho-mechanism does not require large *Dmd* gene rearrangements *Sudmant et al., 2015*. Yet, improvements resulting from the expression of mini/micro dystrophins do not support the requirement for the entire 14 kb transcript or a toxicity of mRNA breakdown products generated via nonsense-mediated decay. However, the impact of premature dystrophin transcript termination on the downregulation of other genes *Kamieniarz-Gdula and Proudfoot, 2019* cannot be excluded.

A more likely explanation involves alterations of epigenetic regulation, a mechanism of increasing importance in DMD: The absence of dystrophin/DAPC in satellite cells leads to aberrant epigenetic activation, which impairs functions of the newly generated cells. This dystrophic SC division defect has been linked to polarization problem due to MARK2 mis-interaction *Chang et al., 2016* combined with β-syntrophin abnormalities *Dumont and Rudnicki, 2016*; *Ribeiro et al., 2019*. Downregulation of β-syntrophin, a DAPC member, results in impaired polarization of its interactor - p38γ. This, in turn, increases CARM1 phosphorylation and reduces *MYF5* activation by PAX7 in in the daughter cell due for myogenic differentiation *Chang et al., 2018*. Furthermore, dystrophin anchors nNOS *Brenman et al., 1995*, which becomes mis-localised in dystrophic cells. This results in decreased nitric oxide-dependent S-nitrosylation of HDAC2 *Colussi et al., 2008* and higher deacetylase activity that has been shown to be involved in the progression of muscular dystrophy *Minetti et al., 2006*.

Thus, loss of the full-length dystrophin in muscle stem cells does not abrogate myogenic cell divisions but causes significant abnormalities. The descendant myoblasts appear to be harbouring somatically heritable epigenetic changes analogous to genomic imprinting *Tucci et al., 2019* and themselves might manifest significant epigenotype abnormalities *Zoghbi et al., 2016*.

Indeed, our *DMD* knockdown and rescue experiments in skeletal muscle progenitor cells *Mournetas et al., 2021* showed that phenotypic changes at day 17 were the consequences of some earlier events that occurred at day 10 and that some abnormalities might not be reproduced nor alleviated after crossing a specific checkpoint.

The data here indicate that epigenetic changes in dystrophic primary myoblasts may be present, as amongst the significantly altered genes *Hist2h2ac*, *Hist1h2ag*, *Hist1h2ah*, *Hist1h1a*, *Smyd1* and *Hmga1* gene products are involved in chromatin modifications or belong to the GO category 'negative regulation of gene expression, epigenetic'. Methylated histone H3 lysine 4 (H3K4) is a key epigenetic signal for gene transcription. H3K4 methylation is mediated by several methyltransferases, including muscle-active *Smyd1 Tan et al., 2006*, found among the top downregulated transcripts in dystrophic myoblasts. Histone methylation can be reversed by histone demethylases, of which Lysine-Specific Histone Demethylase 1 A (LSD1 encoded by *Kdm1a*) is an important muscle enzyme *Tosic et al., 2018*. The network of interactions between genes in this aforementioned GO category (*Appendix 1—figure 3*, *Supplementary file 1-8*) showed a clear interface between the *Kdm1a* node and significantly downregulated histone genes. LSD1 is involved in controlling *Myod1* expression *Scionti et al., 2017*, which we found significantly downregulated in dystrophic myoblasts. Importantly, treatment of *Dmd*<sup>mdx</sup> mice with histone deacetylase inhibitors (HDACi) promoted myogenesis *Saccone et al., 2014*. Epigenetically, HDACi upregulate *Myod1 Mal et al., 2001* and therefore would counteract its decreased expression and the resulting downregulated expressions of genes controlled by this TF, which we described here.

We postulate that, in dystrophic myoblasts, epigenetic dysregulation of *Myod1* expression causes a pathological cascade of downregulated transcriptions of a range of genes controlled by MyoD, with functional consequences for muscle regeneration.

If it is the absence of dystrophin in myogenic cells that determines their fate, developmental muscle abnormalities should occur. This is indeed the case: asymptomatic DMD patients already have transcriptomic alterations *Pescatori et al., 2007* and myogenesis in *Dmd*^mdx embryos is severely disrupted, with myotube hypotrophy, reduced myotube numbers and displacement defects *Merrick et al., 2009*. This developmental abnormality continues in adult muscles during regeneration.

Altogether, our data identify a continuum, where mutations disrupting expression of full-length dystrophin cause SC division abnormalities impacting myoblast generation but also imprinting these cells with functionalities further reducing muscle regeneration. And as dystrophin is needed at the myotube stage for the initial assembly of the DAPC and subsequent formation of viable myofibres *Shoji et al., 2015*, dystrophinopathy results in contraction-induced injury in mature muscle (*Rader et al., 2016*). Thus, the vicious circle of disease is closed with both degenerative and regenerative abnormalities contributing to DMD progression (*Figure 10*).

Such a scenario explains the apparent contradiction, where removal of dystrophin and/or DAPC in fully developed myofibres does not result in dystrophic muscle damage (*Rader et al., 2016*; *Ghahramani Seno et al., 2008*). It would be the case if, once formed in a healthy muscle, DAPC remains stable even if dystrophin is subsequently lost, as indeed have been demonstrated (*Ghahramani Seno et al., 2008*).

Moreover, as the efficacy of exon skipping and gene targeting methods is much higher in proliferating cells than in myofibres, all outcomes of these treatments could arise from dystrophin re-expression in myoblasts, which later differentiate into myotubes with the functional DAPC.

Findings described here agree with the existing data that loss of dystrophin expression disrupts many downstream processes. These processes offer good targets for therapeutic interventions that are not constrained by the causative mutation *Percival et al., 2012*; *Vidal et al., 2012*; *Sinadinos et al., 2015*. Importantly, adjustments made in dystrophic myoblasts can reduce or even prevent occurrence of dystrophy. For example, upregulation of Jagged 1 can counteract damage in muscles without any dystrophin *Vieira et al., 2015* and corrections to COUP-TFII *Xie et al., 2016*, EGFR-Aurka and Ghrelin expression *Wang et al., 2019* can reduce muscle damage.

Given that these pathways are active in myoblasts and the proteins involved are structurally incompatible with substituting the scaffolding properties of dystrophin, these findings further challenge the role of dystrophin in myofibre sarcolemma stabilization as the key pathological alteration in DMD.

Interestingly, the metabolic analysis indicated a significant alteration in key metabolic pathways in DMD myoblasts, which were in line with alterations in the rate of glycolysis/gluconeogenesis, tricarboxylic acid cycle, fatty acid activation, and propanoate metabolism previously found in DMD patients (as previously reviewed *Timpani et al., 2015*). Clearly, the DMD impact on metabolism is an early and widespread disease manifestation, with abnormalities presenting already in the embryonic development *Mournetas et al., 2021* and present in myoblasts *Vallejo-Illarramendi et al., 2014* (and this work), in differentiated muscle and also in dystrophic brains *Kreis et al., 2011*. As an increase in energy demand is observed in DMD muscle, the upregulation of the glycolysis/gluconeogenesis metabolic pathway suggests the upregulation of glucose metabolism-related enzymes, including hexokinase (D-glucose:ATP) and pyruvate kinase, which compensates for the energy demand *Filippelli and Chang, 2021*. These changes are also compensated by other metabolic alterations in mitochondrial pathways and reactions, for example the downregulation of mitochondrial beta-oxidation of branched-chain fatty acids, odd-chain fatty acids, and di-unsaturated fatty acids (n-6) *Nsiah-Sefaa and McKenzie, 2016*. Notably, these metabolic adaptations are activated in dystrophic myoblasts and not just myofibres, which could be easier explained by increased energy demands.

The synthesis of ATP and phosphocreatine is catalysed by creatine kinase in a reversible mechanism *Bradley, 2004*; *Scholte and Busch, 1980*. Therefore, creatine kinase, significantly downregulated in Arginine and proline metabolism of mitochondrial centre (*Figure 9b*) signifies the reduction of ATP production in DMD myoblasts *Kelly-Worden and Thomas, 2014*. The ATP dysfunction at the mitochondrial level induces several patho-physiological events and should exacerbate dystrophinopathy *Scholte and Busch, 1980*.

The mitochondrial leukotriene metabolism is found to be significantly elevated in DMD myoblasts, consistently with the fact that muscle can produce leukotrienes, which are implicated in inflammation and contribute to muscle weakening *Korotkova and Lundberg, 2014*. Furthermore, the observed upregulation in propanoate metabolism leads to the accumulation of propionyl-CoA and associated

metabolites in plasma, and increases the levels of propionyl-CoA and protein propionylation, which can impair myogenic differentiation in human myoblasts *Lagerwaard et al., 2021*.

Glucose-6-phosphate dehydrogenase is known to contribute to the regulation of glucose uptake in skeletal muscle *Lee-Young et al., 2016*. The downregulated fold change of glucose 6-phosphate dehydrogenase (G6PDH2cNo1) suggests that the concentration of glucose is significantly altered in DMD myoblasts. The reduction in the glucose level could impair the differentiation of skeletal myoblasts through the activation of a pathway that targets the enzymatic activity of SIRT1 *Fulco et al., 2008*.

Our findings here, combined with the other data, provide compelling foundation for the call to re-evaluate the established belief on the pathogenesis of DMD. Such reconsideration is important because most current therapeutic approaches aim at dystrophin restoration in differentiated myofibres and are often conducted in the advanced-stage disease, with very limited regenerative muscle potential. Despite decades of intensive research and numerous clinical trials, none of the candidate treatments delivered disease-modifying results *Rodrigues and Yokota, 2018*. Part of the problem is immune responses to re-expressed dystrophin *Vila et al., 2019*, which need to be tackled *Sharma et al., 2017* or prevented. However, dystrophin re-expression using new technologies *Mangeot et al., 2019* allowing re-targeting dystrophin to myoblasts and satellite cells in younger patients or, better still, in newborns able to develop neonatal tolerance to dystrophin, could produce effective therapies for this devastating disease.

## Materials and methods
### Animals
Male C57BL/10ScSn-*Dmd^mdx*/J, C57BL/10ScSnJ, C57BL/6-DmdGt(ROSAbgeo)1Mpd/J (*Dmd^mdx-βgeo*) and C57BL/6 J eight week old mice were used in accordance with institutional Ethical Review Board and Home Office (UK) Approvals. All mice were maintained under pathogen-free conditions and in a controlled environment (12 hr light/dark cycle, 19–23°C ambient temperature, 45–65% humidity). For environmental enrichment, tubes and nesting materials were routinely placed in cages. The genotypes of all experimental animals were confirmed by PCR, as described *Sinadinos et al., 2015*.

The C57BL/10 and C57BL/6 strains derived from a common origin *Petkov et al., 2004* and it has been demonstrated that the mdx mutation on the C57BL/6 background shows the same pathology as the original C57BL/10 strain *McGreevy et al., 2015*. Nevertheless, for RNA-Seq experiments, mixed background animals were generated to facilitate the direct comparison between *Dmd^mdx* and *Dmd^mdx-βgeo* transcriptomes. For this, *Dmd^mdx* males were paired with C57BL/10 females and C57BL/6 males were paired with *Dmd^mdx-βgeo* females. The resulting *Dmd^mdx/WT* females and *Dmd^mdx-βgeo* males were then bred together. The resulting *Dmd^WT* males were used as controls and bred with the *Dmd^mdx/mdx-βgeo* females. The next generation *Dmd^mdx* and *Dmd^mdx-βgeo* males were used as the dystrophic groups. For the subsequent functional pairwise analyses, *Dmd^mdx* were compared against C57BL/10 and *Dmd^mdx-βgeo* against C57BL/6 mice.

### Primary myoblast extraction and culture
Gastrocnemii from 8-week-old male mice (during the active disease period) were used for myoblast extraction and culture. *Gastrocnemius* was chosen as it is considered a fast muscle with predominantly IIB fibres *Augusto et al., 2004* and therefore more prone to dystrophic changes and producing a relatively homogenous population of myoblasts. Briefly, muscles were dissected free of fat and connective tissue before enzymatic digestion by 0.2% type IV collagenase (Sigma, C5138) in Dulbecco's Modified Eagles' Medium (DMEM, Gibco 31885–023) for 90 min at 37 °C, 5% $CO_2$. The digested muscles were then rinsed in plating medium (10% horse serum, Gibco 26050–088; 0.5% chicken embryo extract, Seralab CE-650-J; 100 units penicillin / 0.1 mg streptomycin per mL, Sigma P0781; in DMEM, Gibco 31885–023) incubated at 37 °C, 5% $CO_2$. Digested muscles were then disrupted by successive passages in 50 mL, 25 mL, and 10 mL serological pipettes. The freed muscle fibres were transferred to a dish containing plating medium.

The dish was then placed under a stereo microscope and 200 live fibres were transferred to another dish containing plating medium. This procedure was repeated 2 more times until 150 fibres were transferred to a third dish. 120 live fibres were then transferred to a tube containing proliferating medium (20% foetal bovine serum, Gibco 10500–064; 10% horse serum, Gibco 26050–088; 0.5% chicken embryo extract, Seralab CE-650-J; penicillin / streptomycin, Sigma P0781; in DMEM, Gibco 31885–023). The resulting suspension of muscle fibres was then disrupted by passing it through an 18-gauge needle 10 times before passing the resulting solution through a 40 μm cell strainer (BD Falcon 352340).

The strained cell suspension was transferred to a collagen I (Sigma C8919) coated cell culture dish and proliferating medium was added to obtain an appropriate volume for the dish. The cells were then cultured at 37 °C, 5% $CO_2$ humidified atmosphere and used for assays and experiments. The purity of myoblast cultures was confirmed by enumeration of cells immunofluorescently stained for MyoD (*Appendix 1—figure 4*). Cells were detached from dishes using Accutase (Biowest L0950). Myoblasts were never passaged more than once and cells from the same passage for dystrophic and control genotypes were used for comparabilty.

## Myoblasts cell lines

The SC5 (mdx) and IMO (WT) cell lines were derived from the leg muscle of the H2Kb-tsA58 line *Morgan et al., 1994*. Cells were tested for mycoplasma using MycoAlert kit (Lonza), authenticated by gPCR for the presence or absence of mdx mutation *Sinadinos et al., 2015* and cultured in DMEM supplemented with 20% FCS, 4 mM L-glutamine, 100 unit/ml penicillin, 100 μg/ml streptomycin and 20 unit/ml murine γ-interferon (Invitrogen) at 33 °C, 5% $CO_2$ humidified atmosphere. When cells reached 95% confluence, they were cultured as primary cells in KnockOut DMEM (Invitrogen) supplemented with 10% v/v Knockout Serum Replacement (KSR, Invitrogen), 5% v/v Donor Horse Serum (DHS, Sera Labs) and 2 mM L-glutamine at 37 °C.

## RNA extraction

Myoblasts isolated from gastrocnemii from three individual mice per experimental group were treated as biological replicates. Total RNA was extracted from $Dmd^{mdx}$, $Dmd^{mdx-\beta geo}$ and corresponding wild-type myoblasts using RNEasy Plus Universal mini kits (Qiagen 73404). Briefly, cells were washed 2 times with warm Dulbecco's phosphate buffered saline DPBS, Gibco 14190144 before lysis by applying QIAzol directly to the freshly washed cells. Lysate was then homogenised by passing it through a 25-gauge needle 20 times. Samples were then processed according to the kit manufacturer's instructions. RNA quality and concentration were measured using a NanoDrop 1000 Spectrophotometer (Thermo Scientific). RNA integrity was assessed using electrophoresis of 100 ng of total RNA in a 1% agarose gel (Sigma A4718) in TRIS-acetate-EDTA (TAE) buffer or using an automated capillary electrophoresis system (2100 Bioanalyzer Instrument G2939BA, Agilent) using a kit assay (Agilent RNA 6000 Nano Kit 5067–1511, Agilent).

## RNA Sequencing

Total RNA samples (n=3 per group) were processed by TheragenEtex (Republic of Korea) using an Agilent Bioanalyzer 2100 for quality assessment of samples with a threshold of 7.0 for the RNA integrity number (RIN). An Illumina TruSeq stranded total RNA kit was used to generate libraries following a ribodepletion step using a Ribo-Zero Human/Mouse/Rat kit. Libraries were sequenced in a paired-end 100 bp run using an Illumina HiSeq 2500 sequencing platform.

Raw reads were quality assessed using fastQC *Andrews, 2010* and reads were trimmed using trim-galore *Krueger, 2012* with parameters "`--illumina` -q 20 `--stringency` 5 -e 0.1 `--length` 40 `--trim-n`" to remove adapter sequence and low-quality sequence tails. Reads were mapped to the GRCm38 *Mus musculus* genome from Ensembl using the STAR universal RNA-Seq aligner *Dobin et al., 2013* with parameters "`--outSAMmultNmax` 300 `--outSAMstrandField intronMotif`". Output mapping files were processed and filtered to remove non-mapping reads with mapping quality less than 20 using samtools *Li et al., 2009a*.

The DESeq2 package *Love et al., 2014* in R *R Development Core Team, 2021* was used to perform differential gene expression analysis. Gene models were taken from Ensembl version 91, and read counts over unique genes were quantified using the *summarizeOverlaps*() function in the GenomicAlignments package *Lawrence et al., 2013* using parameters 'mode = "Union", singleEnd = FALSE, ignore.strand = FALSE, fragments = FALSE, preprocess.reads = invertStrand'. p values were adjusted for multiple comparisons by using the *Benjamini and Hochberg, 1995* False Discovery Rate correction. Significantly differentially expressed genes were identified based on a fold-change of two-fold or greater (up- or down-regulated) and an adjusted p-value < 5.0e-2. An additional filter was put in place to remove genes where the mean normalised Fragments Per Kilobase Mapped (FPKM) was <1 for both conditions to avoid changes in low abundance transcripts.

GO enrichment analysis was conducted using the clusterProfiler package *Yu et al., 2012* in R *R Development Core Team, 2021*. REVIGO *Supek et al., 2011* was used to reduce the redundancy of GO enrichment data to rationalise the categories being compared in figures. The default 'SimRel' semantic similarity

measure with a 0.4 threshold cut-off was used. When filtering GO categories for redundancy, the mouse UniProt database was used for all analyses, except when analysing the list of GO categories enriched in both human and mouse primary datasets, in which case the human UniProt database was used. STRING *Szklarczyk et al., 2019* was used to explore and visualize interaction networks. Only interactions present in databases for relevant species or experimentally determined were used.

Regulatory regions of the differentially expressed genes (listed in *Supplementary file 1-1, 1-2, 1-4* and *supplementary file 2*), defined as the region 1Kb upstream or downstream of the transcription start site (TSS) of the gene, were analysed for overrepresentation of binding sites of transcription factors (TFs). The binding signal was analysed using the seqinspector tool (http://seqinspector.cremag.org), *Piechota et al., 2016* and publicly available ChIP-Seq data *ENCODE Project Consortium, 2004*. The overrepresentation was assessed by comparing the lists of tested genes with 1000 randomly selected gene promoters (*Mus musculus* reference genome mm9) Ensembl 75 gene symbols. TF signal was considered enriched if Bonferroni-corrected p-value returned by seqinspector was <5.0e-2.

RNA-Seq data have been deposited in the ArrayExpress database at EMBL-EBI under accession number E-MTAB-10322 for the primary mouse myoblast samples (https://www.ebi.ac.uk/arrayexpress/experiments/E-MTAB-10322) and accession number E-MTAB7287 for SC5 and IMO myoblast cell line samples (https://www.ebi.ac.uk/arrayexpress/experiments/E-MTAB-7287).

## Human primary adult myoblast RNA-Seq data

The collection of primary myoblasts was established as part of the medical diagnostic procedure. A signed and informed consent was obtained by the Cochin Hospital cell bank/Assistance Publique-Hôpitaux de Paris for each patient to collect, establish, and study primary cultures of cells. This collection of myoblasts was declared to (i) legal and ethical authorities at the Ministry of Research (declaration number 701, modified declaration number 701–1) *via* the medical hosting institution (Assistance Publique-Hôpitaux de Paris) and to (ii) the 'Commission Nationale de l'Informatique et des Libertés' (declaration number 1154515). RNA-Seq data from human dystrophic (duplication exons 3–26, deletion exons 8–43, and stop exon 7) and healthy primary myoblasts were generated and analysed as described *Mournetas et al., 2021*. Raw RNA-Seq data have been deposited in the ArrayExpress database at EMBL-EBI under accession number E-MTAB-8321 (https://www.ebi.ac.uk/arrayexpress/experiments/E-MTAB-8321). This RNA-Seq data was then used to analyse the metabolic changes in DMD myoblasts.

The complete fold change values for the human DMD and healthy samples estimated using the DESeq2 package (as described in the RNA Sequencing section) were further used to reconstruct the DMD-specific Genome-Scale Metabolic Model (GSMM), as described below. The genes with NaN fold change were unperturbed for the metabolic analysis.

## Reconstruction of the genome-scale metabolic model

The myoblast-specific metabolic model was constructed using an extensively curated GSMM of human metabolism *Robinson et al., 2020*. As significant enzymatic changes are observed in DMD *Aoyagi et al., 1983*, and in order to fully take into account enzymatic activity, an enzyme-constrained Human model, built with GECKO version 1.0 *Sánchez et al., 2017* was used.

Using the previously developed method for building context-specific models *Angione, 2018*, the fold change value of RNA-Seq data was integrated into GSMM to constrain the myoblast-specific metabolic network *Rawls et al., 2019*. The idea is to constrain the base enzyme constrained human metabolic model with upper and lower bounds on metabolic reactions depending on the fold change value of gene expression data, also taking into account the role of each gene in each reaction *Zampieri et al., 2019*. Based on previously reported increased or decreased enzymatic activities stated in the literature, the upper and lower bound of the metabolic reactions associated with the enzymes were further constrained.

The context-specific model was then solved using flux variability analysis (FVA) to obtain the minimum and maximum flux rate for each reaction, satisfying the given constraints. The FVA was performed by solving a 2 n linear programming problem on the constrained metabolic model using the COBRA toolbox *Heirendt et al., 2019* and the gurobi solver. The full details of the metabolic modelling framework are reported in the Appendix 2.

## cDNA synthesis and qPCR analysis

Total RNA samples were converted to cDNA using SuperScript VILO cDNA Synthesis Kit (Invitrogen 11754050) as per manufacturer instructions.

qPCR reactions were run in duplicates using 25 ng of cDNA per reaction with Precision Plus Mastermix (Primer Design PPLUS-LR), forward and reverse primers synthesized by Eurofins (*Supplementary file 1-9*) and DEPC treated water (Fisher Bioreagents BP561) as per manufacturer instructions, in 96 well plates using an Applied Biosystem ViiA7 RT-PCR instrument, and expression quantified using the ΔΔCT method.

## Cell proliferation assay

Primary myoblasts were seeded in 10 cm diameter dishes (Sarstedt 83.3902) coated with Collagen I (Sigma C8919) at 30% confluency in proliferation medium, as above and left to attach for 2 hr before adding bromodeoxyuridine / 5-bromo-2'-deoxyuridine (BrDU, Invitrogen B23151) to a final concentration of 75 μM and left to proliferate for 6 hr. Cells were then rinsed with warm DPBS, detached using Accutase (Biowest L0950) and fixed using ice cold ethanol. Samples were left at –20 °C over-night and then analysed by flow cytometry. Fixed cells were submitted to an acid wash (2 M HCl, 0.5% Triton X-100) for 30 min at room temperature, washed in PBS and the remaining acid was neutralized using a borate buffer (pH = 8.5). Cells were then stained in 200 μL of PBS (0.5% Tween 20) with 1 μg of Anti-BrDU FITC conjugated antibody (BD 556028) or an FITC conjugated isotype control for 30 min at room temperature. Cells were then washed in HEPES buffer (pH = 7.4) and resuspended in PBS before processing using a FACScalibur system.

## Boyden chamber chemotaxis assay

Primary myoblasts from dystrophic and WT mice were treated for 2 hr with proliferating medium complemented with 10 μg.mL$^{-1}$ mitomycin C (Fisher Bioreagents BP2531) to prevent proliferation from confounding the results of the assay. Cells were then rinsed with warm DPBS twice to remove excess mitomycin C. Each insert (Sarstedt 83.3932.800) was seeded with 10 000 myoblasts in serum-free medium while the wells were filled with either proliferating medium or serum-free medium complemented with 10 ng.mL$^{-1}$ FGF-b (R&D Systems 3139-FB), IGF-I (R&D Systems 791 MG) and HGF (R&D Systems 2207-HG). Cells were left for 12 hr before fixation in buffered formalin (Sigma HT501128) and crystal violet staining for cell counting and spectrophotometer measurements.

## Cell differentiation assay

Myoblasts were used to generate spheroids as such a culture provides an environment that facilitates myogenic differentiation due to a 3D structure improving physical interactions, differentiation and fusion of myoblasts *Chiron et al., 2012*; *Corona et al., 2012*; *Mei et al., 2019*; *Kim et al., 2020*. One million myoblasts per genotype and per mouse were pelleted at 300 g for 5 min in 15 mL centrifuge tubes before pellets were transferred into individual wells of a six-well plate containing 50% proliferation medium (20% foetal bovine serum, Gibco 10500–064; 10% horse serum, Gibco 26050–088; 0.5% chicken embryo extract, Seralab CE-650-J; penicillin / streptomycin, Sigma P0781; in DMEM) and 50% DMEM (Gibco 31885–023) and cultured for 24 hr. Then cells were placed in 25% proliferation medium with 75% DMEM for 24 hr and finally in the differentiation medium (2% horse serum, Gibco 26050–088; penicillin / streptomycin, Sigma P0781; in DMEM). Introduction to differentiation medium was considered t=0 and differentiation was allowed for 6 days.

At the collection time, spheroids undergoing differentiation were rinsed in DPBS and collected in 1 mL of Qiazol solution before freezing for subsequent processing. On thawing, spheroids were disrupted using progressively smaller gauge needles (19, 21, 23, and 25 gauge) and then processed as *per* the RNA extraction section. *Mymk* was chosen as a marker of myoblast fusion and therefore entry into the final stages of differentiation *Chen et al., 2020* and *Myog* and *Myh1* as robust markers of late stage myogenic differentiation *Stern-Straeter et al., 2011*.

## Immunolocalisation with confocal microscopy

Muscles (tibialis anterior) frozen in liquid N$_2$ chilled-isopentane in cryo-embedding matrices (Cellpath KMA-0100–00 A) were allowed to equilibrate to –20 °C in the cryostat chamber. Cryosections (10 μm) were cut and attached to poly-L-lysine-coated slides (Thermo Scientific J2800AMNZ). Sections were allowed to air-dry before fixation and staining. Myoblasts were seeded at 10,000 per collagen-coated coverslip in a 24-well plate (Sarstedt 83.1840.002 and 83.3922) and left to adhere for 12 hr before

processing. Spheroids having undergone the differentiation process were frozen in embedding medium and 10 µm sections were cut, as previously described.

Both muscle sections and cultured myoblasts were then processed identically. They were fixed in 10% neutral buffered formalin (Sigma-Aldrich HT501128) at room temperature for 10 min and washed 3 times in PBS containing 0.1 % v/v Triton X-100 (Sigma-Aldrich X100), blocked for 30 min using 10 % v/v chicken serum (Gibco 16110082) in PBS-Triton then incubated for 90 min under agitation in primary antibody solution: anti dystrophin (Developmental Studies Hybridoma Bank MANDYS1 clone 3B7 cell culture supernatant) 25 % v/v antibody or anti Myosin-4 (Invitrogen, 14-6503-95) 1 µg.mL$^{-1}$, 10 % v/v chicken serum in PBS-Triton. Samples were then washed one time in PBS-Triton then incubated in secondary antibody solution (Invitrogen A-21200) with 10 µg.mL$^{-1}$ antibody, 10 % v/v chicken serum and DAPI 300 nM (Invitrogen D1306) in PBS-Triton for 1 hr followed by two washes in PBS-Triton then PBS and mounted using an anti-fading mounting medium (Thermo Scientific TA-006-FM). Omitting primary antibodies in the reaction sequence served to confirm the specificity of the signal.

Samples were examined with a confocal laser-scanning microscope (LSM 880; Zeiss, Oberkochen, Germany) using a Plan Apochromat, 63 x DIC oil objective (NA 1.4) (pixel size 0.13 µm) objective. Images were acquired using sequential acquisition of the different channels to avoid crosstalk between fluorophores. Pinholes were adjusted to 1.0 Airy unit. In all cases, where multiple images were captured from the same immunohistochemical reaction, laser power, pinhole, and exposure settings were captured once on sample from a representative control section and maintained throughout imaging. Images were processed with the software Zen (Zeiss) and exported into tiff images for figure generation or WMV for video exporting. Only brightness and contrast were adjusted for the whole frame, and no part of any frame was enhanced or modified in any way.

For immunodetection of MyoD, freshly isolated cells were re-plated on coverslips in growth medium and allowed to grow to 70–80% confluence. After rinsing twice with PBS (w/o calcium and magnesium) cells were fixed in a 4% w/v paraformaldehyde solution (PFA) in PBS for 10 min at room temperature (RT), permeabilized in PBS with 0.1% Triton X-100 for 5 min and blocked in a 5% v/v goat serum (GS, Normal Goat Serum, S-1000, Vector Laboratories IVD) in PBS for 1 hr at RT. Cells were incubated with MYOD monoclonal antibody (MA5-12902 ThermoFisher Sci.) diluted 1:100 in the blocking buffer at 4 °C overnight. The secondary antibody (Alexa Fluor 594 goat anti-Mouse, ThermoFisher A11032, 1:1000 in 5% GS in PBS) was applied for 1 h at RT in the dark. Cells were rinsed in PBS and washed twice for 5 min with 1% GC with slow agitation. Then cells were incubated with Hoechst (1:1000 from stock solution 10 mg/ml) for 10 min at RT in the washing buffer. Finally, cells were rinsed and washed two times for 5 min with PBS with agitation. The coverslips were mounted onto microscope slides in Glycergel Mounting Medium (VectaShield, Vector Laboratories) prior to imaging. Images were obtained using a confocal microscope (Zeiss Spinning Disk Confocal Microscope Axio Observer Z.1) and image analysis was performed using ImageJ software.

## Statistical analysis

Student's unpaired t-test was used for comparisons between two data groups. Two-way ANOVA was used to determine if there was any statistically significant difference between groups and time points in the differentiation assay and Fisher's LSD was used to determine which time points exhibited statistically significant differences between genotypes. Pearson's correlation coefficient was calculated to quantify the level of correlation between datasets, and Fisher's exact test was used to further investigate the similarities between human and mouse primary myoblasts gene expression data. All results which passed the threshold for statistical significance were tested for normal distribution using the Shapiro-Wilk normality test, all analysed data were normally distributed. A p-value of ≤5.0e-2 was considered statistically significant, and the values are reported as follows in figures: *p≤5.0e-2, **p<1.0e-2, ***p<1.0e-3, ****p<1.0e-4. All statistical analyses outside of differential gene expression and GO enrichment analyses were performed using GraphPad Prism 8.4.2.

A one-sided hypergeometric test with false discovery rate adjustment for multiple testing on differentially regulated reactions was used to uncover the most perturbed metabolic pathways between the two groups, as described before *Yaneske et al., 2021*. The flux fold change of the DMD-specific max flux and the baseline human max flux was estimated as illustrated in *Equation 4*.

$$fluxfoldchange = \frac{MaxFlux_{DMD}+1}{MaxFlux_{baseline}+1} \qquad (4)$$

where $MaxFlux_{DMD}$ is the Max Flux from DMD-specific GSMM, $MaxFlux_{baseline}$ is the Max Flux from the baseline healthy human GSMM (without any constraints). To avoid 0/0 or NANs ratios, a small correction coefficient +1 was added to both DMD and healthy Max Fluxes. The log2 of flux fold change was used to identify the upregulated and downregulated metabolic reactions and pathways. To select metabolic processes characterised by statistically significant activity changes, reactions with a log2(flux fold change) greater than 1 and a log2(flux fold change) distribution above the 95th percentile were classified as 'upregulated reactions', while the reactions with a log2(flux fold change) less than –0.1 and below the 5th percentile log2(flux fold change) were classified as 'downregulated reactions'.

## Acknowledgements

The authors thank S Arkle, C Crane-Robinson and K Zabłocki for the critical comments on the manuscript. SCR was partially funded through an Expanding Excellence in England (E3) grant from Research England. CA acknowledge a Network Development Award from The Alan Turing Institute, grant number TNDC2-100022.

## Additional information

### Competing interests

Samuel C Robson: SR received royalties from Cancer Research Technology Ltd through University of Cambridge Enterprise Ltd under agreement A13086 for "Chromatin-bound METTL3 m6A methyltransferase facilitates myeloid leukaemogenesis by modulating ribosome stalling". The other authors declare that no competing interests exist.

### Funding

| Funder | Grant reference number | Author |
| --- | --- | --- |
| Alan Turing Institute | TNDC2-100022 | Claudio Angione |
| Research England | Expanding Excellence in England (E3) | Claudio Angione |

The funders had no role in study design, data collection and interpretation, or the decision to submit the work for publication.

### Author contributions

Maxime RF Gosselin, Investigation, Writing – original draft, Writing – review and editing; Virginie Mournetas, Malgorzata Borczyk, Formal analysis, Investigation, Writing – review and editing; Suraj Verma, Formal analysis, Investigation, Writing – original draft; Annalisa Occhipinti, Investigation, Formal analysis; Justyna Róg, Investigation, Writing – original draft; Lukasz Bozycki, Investigation, Visualization; Michal Korostynski, Software, Formal analysis, Supervision, Writing – original draft, Writing – review and editing; Samuel C Robson, Data curation, Formal analysis, Supervision, Investigation, Writing – review and editing; Claudio Angione, Conceptualization, Software, Formal analysis, Supervision, Methodology; Christian Pinset, Formal analysis, Supervision, Writing – review and editing; Dariusz C Gorecki, Conceptualization, Supervision, Funding acquisition, Investigation, Writing – original draft, Project administration, Writing – review and editing

### Author ORCIDs

Maxime RF Gosselin ![ID] http://orcid.org/0000-0002-8916-0953
Virginie Mournetas ![ID] http://orcid.org/0000-0002-6557-4190
Samuel C Robson ![ID] http://orcid.org/0000-0001-5702-9160
Claudio Angione ![ID] http://orcid.org/0000-0002-3140-7909
Dariusz C Gorecki ![ID] http://orcid.org/0000-0003-3584-1654

### Ethics

Mice were maintained and used in accordance with institutional Ethical Review Board and Home Office (UK) Approvals. The collection of primary myoblasts was established as part of the medical

diagnostic procedure. A signed and informed consent was obtained by the Cochin Hospital cell bank/ Assistance Publique-Hôpitaux de Paris for each patient to collect, establish, and study primary cultures of cells. This collection of myoblasts was declared to (i) legal and ethical authorities at the Ministry of Research (declaration number 701, modified declaration number 701-1) via the medical hosting institution (Assistance Publique-Hôpitaux de Paris) and to (ii) the 'Commission Nationale de l'Informatique et des Libertés' (declaration number 1154515).

## Decision letter and Author response

Decision letter https://doi.org/10.7554/eLife.75521.sa1
Author response https://doi.org/10.7554/eLife.75521.sa2

## Additional files

### Supplementary files

• Supplementary file 1. Includes 9 tables detailing the bioinformatics analyses. 1–1: Genes significantly dysregulated in $Dmd^{mdx}$ vs. WT primary myoblasts. From left to right, columns show ENSEMBL gene id, gene symbol, log2 fold change between $Dmd^{mdx}$ and WT primary myoblasts, adjusted $p$-value as output by DESeq2, average FPKM for $Dmd^{mdx}$ and WT groups. Genes are ordered from most downregulated to most upregulated. 1–2: Tabulation of genes significantly dysregulated in $Dmd^{mdx-\beta geo}$ vs. WT primary myoblasts. From left to right, columns show ENSEMBL gene id, gene symbol, log2 fold change between $Dmd^{mdx-\beta geo}$ and WT primary myoblasts, adjusted $p$-value as output by DESeq2, average FPKM for $Dmd^{mdx-\beta geo}$ and WT groups. Genes are ordered from most downregulated to most upregulated. 1–3: GO categories significantly enriched in the downregulated gene lists from primary $Dmd^{mdx}$ vs. WT and $Dmd^{md-\beta geo}$ vs. WT. Columns A and D show the GO category ID, columns B and E show GO category description and columns C and F show adjusted $p$-value for GO categories enriched in the significantly downregulated genes in primary $Dmd^{mdx}$ and $Dmd^{mdx-\beta geo}$ myoblasts, respectively. GO categories are ordered from the lowest (top) to the highest (bottom) $p$-value for each analysis and GO categories significantly enriched in both analyses are highlighted in red. 1–4: Genes significantly dysregulated in SC5 (dystrophic) vs. IMO (WT) myoblast cell line. From left to right, columns show ENSEMBL gene id, gene symbol, log2 fold change between SC5 and IMO primary myoblasts, adjusted $p$-value as outputted by DESeq2, average FPKM for SC5 and IMO groups. Genes are ordered from most downregulated to most upregulated. 1–5: GO categories significantly enriched in the downregulated gene lists from primary $Dmd^{mdx}$ vs. WT and SC5 vs. IMO cell lines. Columns A and D show the GO category ID, columns B and E show GO description and columns C and F show adjusted $p$-value for GO categories enriched in the significantly downregulated genes in SC5 myoblast cell line and primary $Dmd^{mdx}$ myoblasts, respectively. GO categories are ordered from the lowest (top) to the highest (bottom) $p$-value for each analysis and GO categories significantly enriched in both analyses are highlighted in red. 1–6: Genes significantly dysregulated in DMD vs. healthy primary human myoblasts. From left to right, columns show ENSEMBL gene id, gene symbol, log2 fold change between DMD and healthy human primary myoblasts, adjusted $p$-value as outputted by DESeq2, average FPKM for human DMD and health groups. Genes are ordered from most downregulated to most upregulated. 1–7: GO categories significantly enriched in the downregulated gene lists from dystrophic vs. WT human primary myoblasts and $Dmd^{mdx}$ vs. WT mouse primary myoblasts. Columns A and D show GO category ID, columns B and E show GO category description and columns C and F show adjusted $p$-values for GO categories enriched in the significantly downregulated genes in primary human DMD and mouse $Dmd^{mdx}$ myoblasts, respectively. GO categories are ordered from the lowest (top) to the highest (bottom) $p$-value for each analysis and GO categories significantly enriched in both analyses are highlighted in red. 1–8: Interactions between genes in the GO category GO:0045814: "Negative regulation of gene expression, epigenetic" generated in STRING. Columns A and B show nodes connected by an edge, column C shows the interaction score if one has been determined experimentally, column D shows the interaction score if an interaction is present in a database feeding into STRING and column E shows the combined interaction score. 1–9: Primers used in real-time quantitative PCR analysis. Primers were designed specifically except *Gapdh* primer set, which was taken from *Otto et al., 2017*.

• Supplementary file 2. Includes the tables detailing the identification of putative molecular regulators of genes altered in $Dmd^{mdx}$ myoblasts. The overrepresentation of binding sites for transcriptional regulators in the promoter region of the altered transcripts was identified using the seqinspector online resource (see Materials and methods). 2 A TFBS analysis results for transcripts

upregulated in *Dmd*<sup>mdx</sup> myoblasts. Column legend: id - ChIP-Seq track id; name - TF name, followed by the index of ChIP-Seq track for this TF; p-value - t-test p-value for overrepresentation of ChIP-Seq signal; Bonferroni - Bonferroni-corrected p-value; foldchange - a ratio of ChIP-Seq signal averaged across all promoters in the gene list versus 1000 random gene promoters; backgroundmean - averaged signal from the track across 1000 random gene promoters; querymean - averaged signal across all promoters from the submitted list. 2-B. TFBS analysis results for transcripts downregulated in Dmd<sup>mdx</sup> myoblasts. columns - as in A. 2 C top 10 genes for each of the significantly overrepresented TFs for transcripts downregulated in Dmd<sup>mdx</sup>. 2-D to 2 G TFBS analysis results (column legend as in A) for the following gene lists: upregulated in Dmd<sup>mdx-βgeo</sup>; downregulated in Dmd<sup>mdx-βgeo</sup>; upregulated in SC-5; downregulated in SC-5. 2 H gene lists as submitted to the seqinspector tool, including gene symbols that were translated for database compatibility.

- Supplementary file 3. Includes tables detailing the transcriptomic and enzyme-constrained flux rate of metabolic pathways and processes, and enrichment analysis results.

- Supplementary file 4. Includes the table detailing the full name of reactions presented in the metabolic map views (*Figure 9* and *Figure 9—figure supplement 1*).

- Transparent reporting form

### Data availability

All data generated or analysed during this study are included in the manuscript and supporting files. Supplementary tables uploaded with the manuscript provide the numerical data that are represented in graphical format in the manuscript figures or as summary tables. RNA-Seq data for the mouse myoblast samples have been deposited in the ArrayExpress database at EMBL-EBI under accession number E-MTAB-10322 and E-MTAB-7287 for primary myoblasts and established cell lines respectively (RNA Sequencing section), and under E-MTAB-8321 for human primary myoblasts (Human primary adult myoblast RNA-Seq data section).

The following dataset was generated:

| Author(s) | Year | Dataset title | Dataset URL | Database and Identifier |
|---|---|---|---|---|
| Górecki D, Gosselin M, Robson S | 2022 | Loss of full-length dystrophin expression causes major cell-autonomous abnormalities in proliferating myoblasts | https://www.ebi.ac.uk/arrayexpress/experiments/E-MTAB-10322 | ArrayExpress, E-MTAB-10322 |

The following previously published datasets were used:

| Author(s) | Year | Dataset title | Dataset URL | Database and Identifier |
|---|---|---|---|---|
| Górecki DC, Robson S | 2019 | RNA-seq comparison of immortalised myoblasts from dystrophic and WT mice | https://www.ebi.ac.uk/arrayexpress/experiments/E-MTAB-7287 | ArrayExpress, E-MTAB-7287 |
| Mournetas V | 2021 | Human primary adult myoblast RNA-Seq | https://www.ebi.ac.uk/arrayexpress/experiments/E-MTAB-8321/ | ArrayExpress, E-MTAB-8321 |

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

## Appendix 1

### Supplementary figures

**Appendix 1—figure 1.** Decreased expression of Dp71 transcript in primary myoblasts from *Dmd*<sup>mdx</sup> mice. Results of qPCR expression assay shown as normalised individual datapoints for biological replicates relative to wild type, normalised to *Gapdh*. Error bars represent mean ± SEM, n=3, *=$p < 5.0e\text{-}2$ (Student's unpaired t-test).

Dystrophin / DAPI

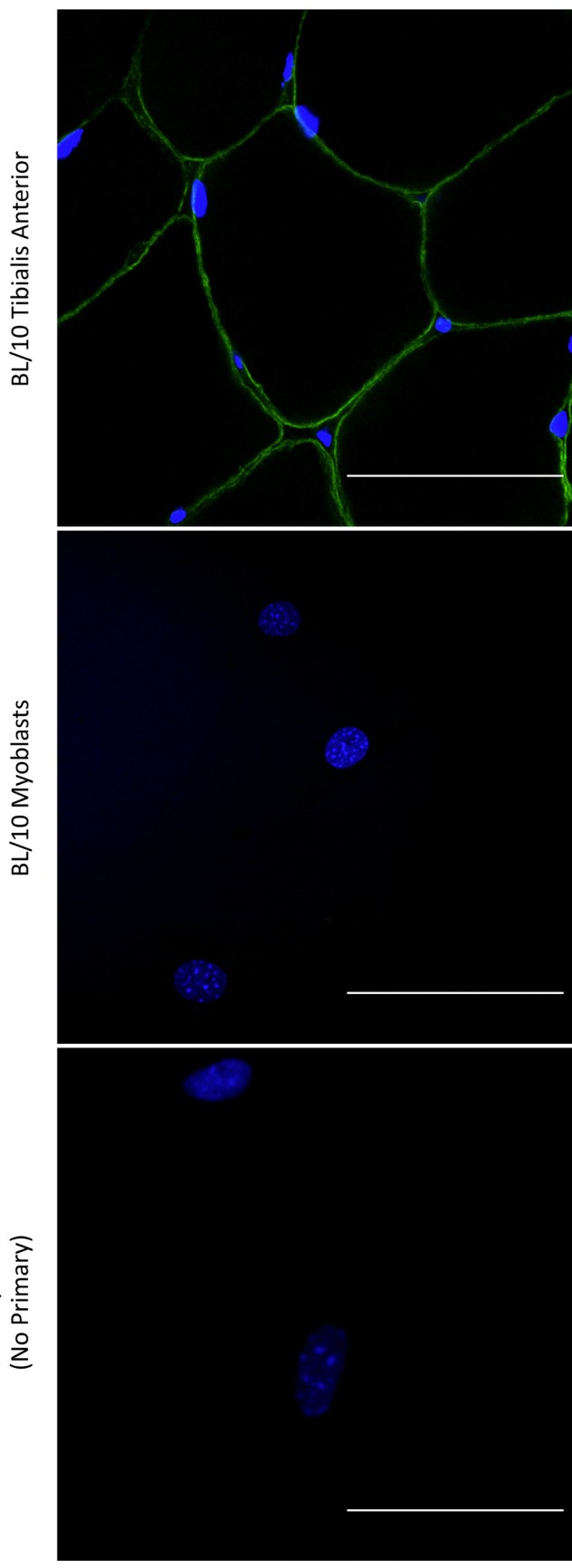

**Appendix 1—figure 2.** Immunodetection of Dp427 in WT primary mouse myoblasts. Representative immunofluorescence micrographs of sections across C57BL/10 *tibialis anterior* muscle and primary *Dmd*mdx myoblasts stained with an antibody detecting epitopes in exons 31/32 of Dp427. Note the expected immunofluorescence signal (green) under the sarcolemma of myofibers and absence of any discernible dystrophin immunoreactivity in myoblasts in culture, reacted and imaged under conditions identical to those used for muscle sections. Negative (no primary antibody) control shown to indicate the signal specificity. Cell nuclei stained with DAPI (blue). Scale bar = 50 μm.

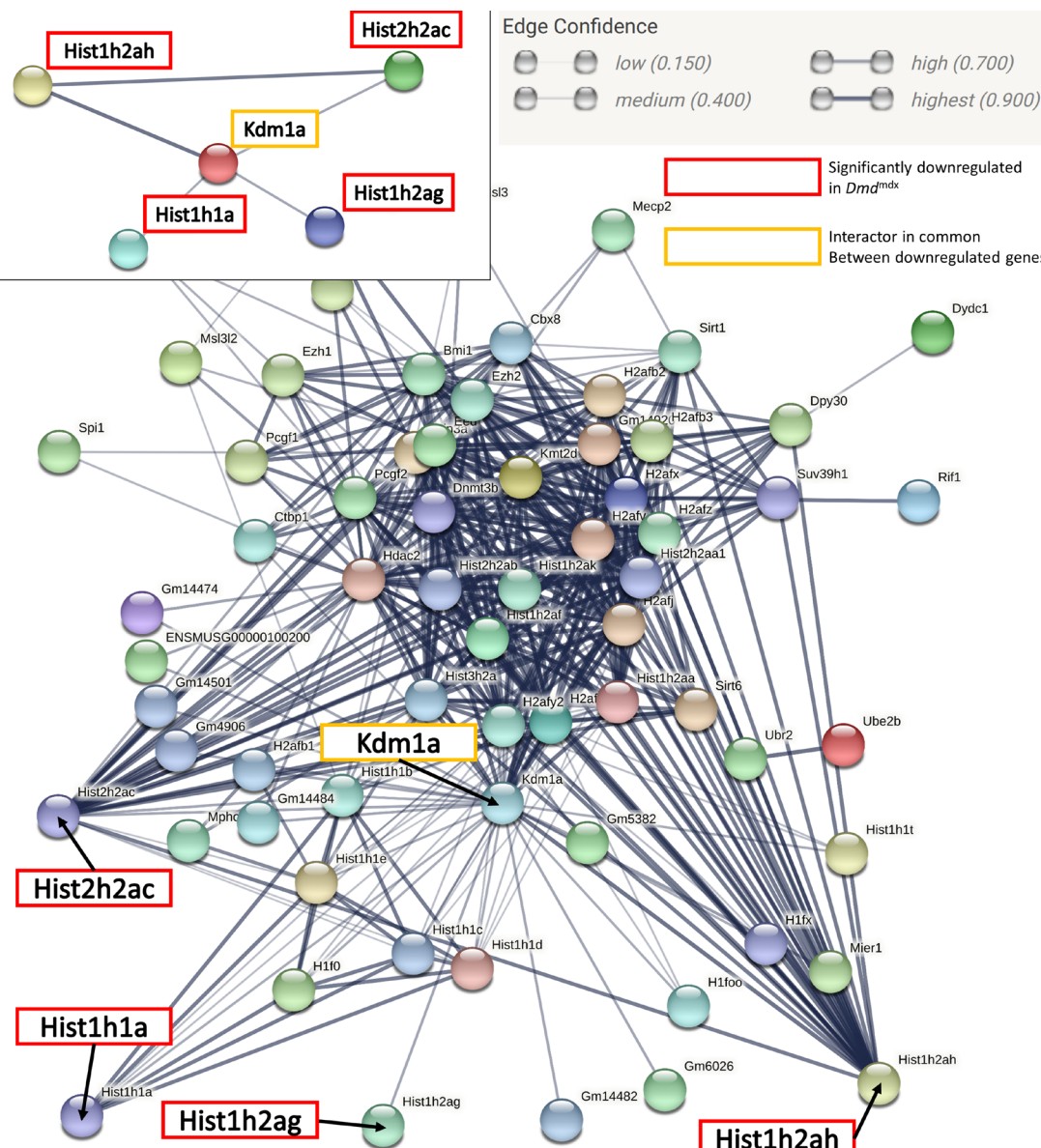

**Appendix 1—figure 3.** STRING interaction diagram of genes in the GO category: Negative regulation of gene expression, epigenetic. Interaction network representing genes in GO:0045814. Each node depicts a gene; edges between nodes indicate interactions between protein products of the corresponding genes and the aggregated interaction scores from experimental data and interaction databases (see the Edge Confidence graphic legend for details). Genes significantly downregulated in dystrophic myoblasts are highlighted in red and the common interactor between all significantly downregulated genes (Kdma1) is highlighted in yellow. Isolated nodes are not shown.

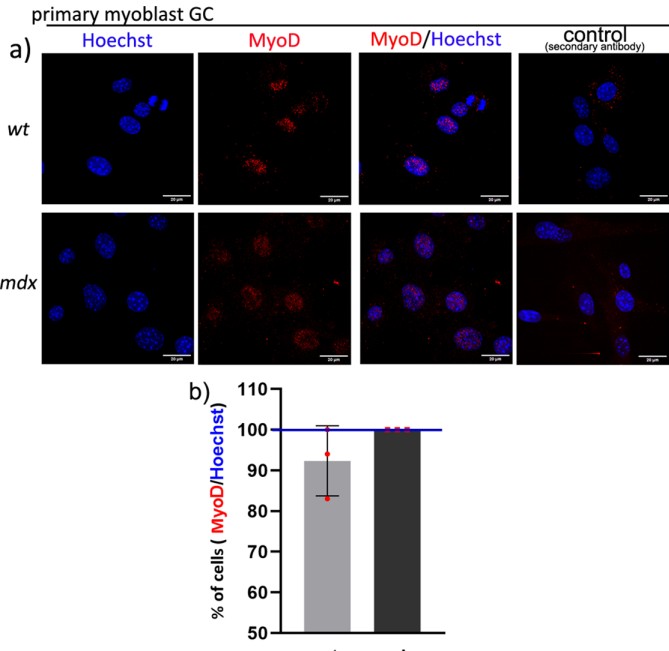

**Appendix 1—figure 4.** The purity of myoblasts isolated from the *gastrocnemius* muscle. (**a**) Example MyoD staining (red signal) and Hoechst nuclear labelling (blue) in primary myoblast cultures isolated from mdx and wild-type gastrocnemii. Representative immunofluorescent image from three tiles are presented and the negative control (no primary antibody) is also shown. (Size bar = 20 µm). (**b**) Graph showing percentage of cells positive for MyoD from the total number of cells identified by the nuclei labelling with Hoechst enumerated using the counter cells function in the ImageJ software. The purity of wt and mdx myoblast cultures was between 93% and 100%. Error bars represent the mean number of cells with SD for three tile images and the difference is not statistically significant.

## Appendix 2

## Genome-scale metabolic model reconstruction methodology

Significant enzymatic and downstream metabolic changes have been previously observed in DMD (*Aoyagi et al., 1983*; *Dreyfus et al., 1956*; *Ragusa et al., 1997*; *Srivastava et al., 2020*). In order to fully take into account both enzymatic and metabolic activity within DMD myoblasts, a condition-specific and enzyme-constrained human metabolic model was constructed to characterise such changes at the genome-scale. As a first step, the Human1 genome-scale metabolic model (GSMM) built with GECKO (*Robinson et al., 2020*; *Sánchez et al., 2017*; *Zampieri et al., 2019*) was retrieved from (https://github.com/SysBioChalmers/ecModels/tree/main/ecHumanGEM/model; *Domenzain et al., 2020*). GECKO reconstructs the GSMM with enzymatic constraints using kinetic and omic data. This is accomplished by expanding the GSMM's stoichiometric matrix to include rows representing enzymes and columns representing enzyme consumption in reactions, whilst enzyme kinetics ($k_{cat}$ values) are depicted in this matrix by pseudo-stoichiometric coefficients. The enzyme usage in the model is constrained as an upper bound of the respective enzyme (protein) reaction, while the lower bound of those reactions is set to 0. Constraining protein abundance lowers flux variability and improves prediction accuracy (*Zampieri et al., 2019*).

Flux balance analysis (FBA) is a mathematical method for analysing the flux of reactions within GSMMs. A GSMM is represented as a stoichiometric matrix *S*, with rows corresponding to metabolites and columns representing reactions. At a steady-state, it is assumed that there is no net change in mass/concentration in the system and that mass/concentration is preserved. As a result, the rate of production of each internal metabolite equals the rate of consumption. A column vector *v* reflects the flow through the system (flux rate of each reaction). Under the steady-state assumption, matrix multiplication of the stoichiometric matrix S and column vector v yields the linear equations reflecting the restrictions (*Sv = 0*), as illustrated in *Equation 1*,

$$
\begin{aligned}
&\max_{v} c^T v \\
&\text{subject to } Sv = 0 \\
&V_{min} \leq v \leq V_{max}
\end{aligned}
\tag{1}
$$

where *S* is a stoichiometric matrix of all known metabolic reactions (metabolites by reactions) and *v* is the vector of reaction flux rates. Additionally, every reaction flux is constrained by lower and upper bounds ($V_{min}$ and $V_{max}$). The linear objective function is represented by the vector c.

Flux Variability Analysis (FVA) is an extended version of FBA, where a range of viable fluxes (i.e. minimum and maximum value) is calculated for each reaction. FVA, therefore, allows exploring the metabolic range of the cell and does not require an objective function, as it iterates over all the reactions as objectives, calculating their minimum and maximum allowable flux (*Rawls et al., 2019*).

Let the vector selecting each biological flux as objective be denoted by *w*. FVA solves two optimization problems (*Equation 2*) for each flux $v_i$ of interest, after solving *Equation 1* with *c=w*.

$$
\begin{aligned}
&\max/\min v_i \\
&\text{subject to } Sv = 0 \\
&w^T v \geq \gamma Z_0 \\
&V_{min} \leq v \leq V_{max}
\end{aligned}
\tag{2}
$$

where $Z_0 = w^T v$ is an optimal solution to *Equation 1*, and γ is a parameter that controls whether the analysis is performed with respect to suboptimal network states ($0 \leq \gamma \leq 1$) (*Gudmundsson and Thiele, 2010*).

The fold change value of RNA-Seq data obtained using DeSeq2 (as described in Section RNA Sequencing), was used to further constrain the GSMM. Using a previous method for building context-specific models (*Angione, 2018*), the RNA-Seq data was integrated into the GSMM to obtain a myoblast-specific metabolic network (*Rawls et al., 2019*). The idea is to constrain the base enzyme constrained human metabolic model with upper and lower bounds on metabolic reactions depending on the fold change value of gene expression data, as illustrated in *Equation 3*.

$$lb\,(i) = lb\,(i) * \left( rxnExprFC\,(i)^{\theta} \right)$$
$$ub\,(i) = ub\,(i) * \left( rxnExprFC\,(i)^{\theta} \right)$$

(3)

where $lb$ is the lower bound and $ub$ is the upper bound of the reaction, $i$ is the index of the reaction, $rxnExprFC$ represents a fold change of each reaction, calculated using the fold change values of the gene expression and taking into account the role of each gene in each reaction (*Vijayakumar et al., 2018*; *Angione and Lió, 2015*). $\theta$ is the hyperparameter used to constrain the model based on the gene expression fold change value. Here, $\theta$ was set to 2.

In addition to the transcriptional constraints described above, we implemented additional constraints based on the literature. Significant enzymatic and metabolic disruption has been shown in DMD, where modest enzymatic alternations may be observed in the early stages but become more extensive with the progression of muscle tissue degeneration (*Dabaj et al., 2018*; *Lindsay et al., 2019*). A significant upregulation in glycolytic enzymes, including Hexokinase-1 and Pyruvate Kinase M2, was observed in dysfunctional muscles, indicating the increased glycolytic activity (*Pant et al., 2015*). The activity of individual glycolytic enzymes (such as alpha-glucan phosphorylase, phosphoglucomutase, and aldolase), creatine kinase and Adenylate kinase 1 is reduced in DMD (*Srivastava et al., 2020*). Furthermore, the first two enzymes of the pentose phosphate pathway of glucose utilisation, glucose-6-phosphate dehydrogenase and 6-phosphogluconate dehydrogenase, have enhanced activity. The activities of lysosomal cathepsin enzymes (such as cathepsins D, A, B1, C, and dipeptidyl peptidase II; protein hydrolyzing enzymes) are also enhanced in muscular dystrophies. Based on these increased or decreased enzymatic activities from the available literature on metabolic alterations, the upper and lower bound of the specific metabolic reactions associated with these enzymes were further constrained as follows.

Firstly, FBA was computed for all reactions using biomass as an objective function. Then the flux rate calculated for the reactions associated with these enzymes was used as a value for the upper bound to ensure reduced enzymatic activity, or as the lower bound to ensure enhanced enzymatic activity, following the cases above. Furthermore, we constrained the biomass output to at least 50% of its maximum growth.

