## [Editor Report]

This is an in depth analysis of the transcriptomic changes occurring in mouse and human myogenic cells that lack of dystrophin. The alterations have implications for the pathogenesis of Duchenne muscular dystrophy. The findings could lead to new therapeutic interventions directed at this highly disabling and lethal disease.

---

## [Decision Letter]

**Decision letter after peer review:**

Thank you for submitting your article "Loss of full-length dystrophin expression results in major cell-autonomous abnormalities in proliferating myoblasts" for consideration by *eLife*. Your article has been reviewed by 2 peer reviewers, and the evaluation has been overseen by a Reviewing Editor and Mone Zaidi as the Senior Editor. The reviewers have opted to remain anonymous.

Essential revisions:

The common view appears to be that whilst recognizing that the RNA seq analysis was well done and may prompt more profound mechanistic study, both reviewers feel that at this stage, the manuscript is a descriptive transcriptional analysis of cell lines isolated from DMD animal models and patients. There is a lack of functional experiments. Furthermore, the phenotypic changes have been evaluated in mouse myoblasts only (not in human). As a result, the proposed putative mechanisms of the paper in its present form are based mainly on the discussion of literature evidence, that can be linked with the transcriptomic data.

To clarify further:

Reviewer 1 felt that your RNAseq-based introductory work could pave the road for a more profound mechanistic study. However, at this stage, the manuscript is a descriptive transcriptional analysis of cell lines isolated from DMD animal models and patients. The authors attempted to explain the observed in vitro cell phenotypes (alteration in proliferation, migration and differentiation) based on several groups of differently expressed genes without any functional analyses. It is hard to predict whether the conclusions have any relevant connection with DMD pathology.

Reviewer 2 felt that The RNA seq analysis was well done. However, the attempts to explain the phenotypic alteration of DMD myoblasts (proliferation, migration and differentiation) lack of functional experiments. Furthermore, the phenotypic changes have been evaluated in mouse myoblasts only (not in human). The proposed putative mechanisms are based mainly on the discussion of literature evidence, that can be linked with the transcriptomic data.

Both reviewers thus feel that the manuscript requires and will greatly benefit from functional experiments aiming at validate/refute the suggested hypothesis.

*Reviewer #1 (Recommendations for the authors):*

At lane 49-50, authors write: "Several studies showed that the ablation of dystrophin in fully differentiated myofibres did not trigger their degeneration 5,6, and even that myofibres can function entirely without dystrophin 7,8.

This is not exactly what reported from the cited refs. For example, in ref.6, the effect of the down regulation of dystrophin was evaluated in the tibialis anterior muscle only, and the authors suggest that: "the myofibres which do not express dystrophin in this model may not be exposed to the usual pressures and forces they are in a dystrophic animal with general weakness."

The results of ref 7 were proved unrepeatable in a subsequent work (Spinazzola JM, Lambert MR, Gibbs DE, et al. Effect of serotonin modulation on dystrophin-deficient zebrafish. Biol Open. 2020;9(8):bio053363.)

Finally, the claim that "myofibers can function entirely without dystrophin" cannot be based on the results presented in ref 8 because the myofibers of the escaper dogs, overexpressing Jagged-1, "showed typical dystrophic features as evidenced by cycles of degeneration and regeneration, which is not seen in normal

muscle".

Lane 426: authors write: "Such a scenario explains the apparent contradiction, where removal of dystrophin and/or DAPC in fully developed myofibres does not result in dystrophic muscle damage 5,6. It would be the case if, once formed in a healthy muscle, DAPC remains stable even if dystrophin is subsequently lost, as indeed have been demonstrated 6".

However, in ref.6 authors did not show that DAPC remains stable after the downregulation of dystrophin: the loss of dystrophin results in the concomitant down-regulation of β-dystroglycan and α-sarcoglycan. In ref.5, the down-regulation of dystroglycan causes an increased susceptibility to contraction-induced injury, albeit in the absence of necrosis.

Lane 436-441: the authors cite references (8,79,80) to support the hypothesis that increased susceptibility of the membrane to rupture caused by dystrophin deficiency would not be the key pathogenic mechanism in DMD. But the cited references showed that increasing the proliferation and differentiation capabilities of satellite cells could increase the regeneration of muscle fibers. Obviously, DMD is a complex disease resulting from increased fiber degeneration and impaired regeneration. That modulation of the regenerative potential of satellite cells may counteract the effect of fiber degeneration does not imply the statement that "these findings further challenge the role of dystrophin in stabilizing myofiber sarcolemma as a key pathological alteration in DMD".

Authors should indicate the source of myoblasts from DMD subjects (biopsies or cell biobank) and the methods used for their purification. Furthermore, they must provide ethic statement for the use of human samples.*Reviewer #2 (Recommendations for the authors):*

The authors extensively discussed many potentially affected molecular pathways in the DMD myoblasts as reasons for the observed alteration in proliferation, migration, and differentiation. A mechanistic study of one or more pathways will significantly increase the manuscript's merit.

---

## [Author Response]

Essential revisions:The common view appears to be that whilst recognizing that the RNA seq analysis was well done and may prompt more profound mechanistic study, both reviewers feel that at this stage, the manuscript is a descriptive transcriptional analysis of cell lines isolated from DMD animal models and patients. There is a lack of functional experiments. Furthermore, the phenotypic changes have been evaluated in mouse myoblasts only (not in human). As a result, the proposed putative mechanisms of the paper in its present form are based mainly on the discussion of literature evidence, that can be linked with the transcriptomic data.

We find the comment on the lack of functional experiments really surprising. We have analysed functionally the key pathways (proliferation, migration and differentiation) essential for myoblast to be able to regenerate muscle, thus proving transcriptomic alterations to be functional and biologically relevant. Our data showing accelerated differentiation in mdx mouse cells fully agree with findings by others, both in primary cultures and in isolated myofibers (Yablonka-Reuveni andAnderson, 2005). We agree that it is essential to link the functional alterations in human dystrophic myoblasts to the transcriptomic alteration that we identified. However, as we explained in the public part of the critique, altered proliferation, migration and differentiation of human DMD myoblasts have been noticed and described before (Witkowski and Dubovitz., 1985; Nesmith et al., 2016; Sun et al., 2020). Our data provide a molecular underpinning for these previously reported abnormalities. Further functional analyses will help to understand their consequences. But it would require investigation of numerous parameters, including the significant alteration in specific metabolic pathways, which we identified in human DMD myoblasts and described in the revised version of this manuscript. Given the convergent yet heterogeneous phenotypes in myoblasts from DMD patients (Choi et al., 2016), such functional analyses would need to involve a significant number of probands.

Therefore, a detailed study in a sufficiently large cohort of DMD myoblasts is a logical next step from the identification of specific pathway alterations described here. But it is an extensive new project beyond our immediate capability.

We believe that the novelty and importance our data, as it is, relies on demonstrating for the first time at the molecular level that the loss of full-length dystrophin expression is both necessary and sufficient to trigger abnormality in myoblasts. Therefore, these findings demonstrate, again for the first time, the pathological vicious cycle, where all muscle cells (satellite cells, myoblasts and myofibers) suffer from the loss of full-length dystrophin expression. We also identified the key mechanisms behind these abnormalities, which stems from the decreased expression of MyoD myogenic factor. Furthermore, we demonstrated that short dystrophin isoforms, although expressed in myoblasts, do not exacerbate the phenotype significantly, if lost. This finding contributes to our understanding of the pathology in dystrophin-null patients.

To clarify further:Reviewer 1 felt that your RNAseq-based introductory work could pave the road for a more profound mechanistic study. However, at this stage, the manuscript is a descriptive transcriptional analysis of cell lines isolated from DMD animal models and patients. The authors attempted to explain the observed in vitro cell phenotypes (alteration in proliferation, migration and differentiation) based on several groups of differently expressed genes without any functional analyses. It is hard to predict whether the conclusions have any relevant connection with DMD pathology.

In fact, the key pathways found altered in RNAseq experiments were then confirmed functionally. In the mechanistic, functional analyses we focussed on proliferation, migration and differentiation as processes known to impact muscle regeneration, thus important for the DMD pathology. In an approach considered as one of the strengths of our work by the other Reviewer, the key findings in primary myoblasts were then reproduced in myoblast cell line, to demonstrate that alterations observed are not evoked by the exposure to the niche environment present in the dystrophic muscle, but that are cell-autonomous. Given the identification of decreased expression of MyoD, which explains the downregulation of other transcripts, as their overwhelming majority is controlled by MyoD, we present the mechanism of these alteration. The mechanistic link between the loss of full-length dystrophin expression and MyoD must now be investigated. But such a study is beyond a reasonable expectation for the experimental work done in this revision.

Reviewer 2 felt that The RNA seq analysis was well done. However, the attempts to explain the phenotypic alteration of DMD myoblasts (proliferation, migration and differentiation) lack of functional experiments. Furthermore, the phenotypic changes have been evaluated in mouse myoblasts only (not in human). The proposed putative mechanisms are based mainly on the discussion of literature evidence, that can be linked with the transcriptomic data.Both reviewers thus feel that the manuscript requires and will greatly benefit from functional experiments aiming at validate/refute the suggested hypothesis.

Our hypothesis was that los of the full-length dystrophin expression causes abnormalities in primary myoblasts and not just in myofibres. We demonstrated this to be the case using a combination of molecular and functional analyses. Contrary to the Reviewer’s opinions, the transcriptomic identification of pathway alterations was just a starting point for our investigations. We are fully aware that pathway alterations are not directional, as up and down regulations of different transcripts in each pathway may result in unpredictable functional changes, and sometimes compensate each other. Therefore, we specifically investigated the key pathways (proliferation, migration and differentiation) in functional experiments. So, it is not a manuscript “based on several groups of differently expressed genes without any functional analyses “.

As for the human myoblasts, other than the expectation that we should do the same experiments in human dystrophic myoblasts, and such experiments were performed and results were described in the literature, the Reviewers are not indicating what functional experiments would be necessary to convince them that the loss of full-length dystrophin expression results in major cell-autonomous abnormalities in proliferating myoblasts.

Reviewer #1 (Recommendations for the authors):At lane 49-50, authors write: "Several studies showed that the ablation of dystrophin in fully differentiated myofibres did not trigger their degeneration 5,6, and even that myofibres can function entirely without dystrophin 7,8.This is not exactly what reported from the cited refs. For example, in ref.6, the effect of the down regulation of dystrophin was evaluated in the tibialis anterior muscle only, and the authors suggest that: "the myofibres which do not express dystrophin in this model may not be exposed to the usual pressures and forces they are in a dystrophic animal with general weakness."The results of ref 7 were proved unrepeatable in a subsequent work (Spinazzola JM, Lambert MR, Gibbs DE, et al. Effect of serotonin modulation on dystrophin-deficient zebrafish. Biol Open. 2020;9(8):bio053363.)Finally, the claim that "myofibers can function entirely without dystrophin" cannot be based on the results presented in ref 8 because the myofibers of the escaper dogs, overexpressing Jagged-1, "showed typical dystrophic features as evidenced by cycles of degeneration and regeneration, which is not seen in normalmuscle".Lane 426: authors write: "Such a scenario explains the apparent contradiction, where removal of dystrophin and/or DAPC in fully developed myofibres does not result in dystrophic muscle damage 5,6. It would be the case if, once formed in a healthy muscle, DAPC remains stable even if dystrophin is subsequently lost, as indeed have been demonstrated 6".However, in ref.6 authors did not show that DAPC remains stable after the downregulation of dystrophin: the loss of dystrophin results in the concomitant down-regulation of β-dystroglycan and α-sarcoglycan. In ref.5, the down-regulation of dystroglycan causes an increased susceptibility to contraction-induced injury, albeit in the absence of necrosis.Lane 436-441: the authors cite references (8,79,80) to support the hypothesis that increased susceptibility of the membrane to rupture caused by dystrophin deficiency would not be the key pathogenic mechanism in DMD. But the cited references showed that increasing the proliferation and differentiation capabilities of satellite cells could increase the regeneration of muscle fibers. Obviously, DMD is a complex disease resulting from increased fiber degeneration and impaired regeneration. That modulation of the regenerative potential of satellite cells may counteract the effect of fiber degeneration does not imply the statement that "these findings further challenge the role of dystrophin in stabilizing myofiber sarcolemma as a key pathological alteration in DMD".

This Reviewer appears to take a particular issue with our Introduction and Discussion sections examining the current dogma on the key significance of dystrophin expression in myofibers and that it might not be entirely correct. This is indeed a key point; therefore, we wish to clarify it for the Editors and the readers. Firstly, we are not suggesting that dystrophin in myofibres is unimportant but that it is important during differentiation (Ref 4) and becomes more redundant once the myofiber is fully differentiated (Ref 5 and 6). The fundamental point is that, contrary to the established belief, the key dystrophin function may not be to provide sarcolemma stability in myofibres but rather that there is a disease continuum: DMD defects in satellite cells cause myoblast dysfunctions diminishing muscle regeneration, and also impair myofiber formation, which subsequently degenerates. We illustrate this notion by our data in this paper and several publications from other laboratories. Rader et al., (Ref 5) in particular, made a similar suggestion in their discussion. We believe that all these data should be considered carefully, because targeting abnormalities in myogenic cells rather than in myotubes may result in effective treatments for this lethal disease. In contrast, attempts at re-expression of dystrophin in myofibres, despite a decade of efforts, failed to provide any disease modifying treatment.

Given the importance of these findings, we have carefully reviewed all points raised and believe that we have not misinterpreted any published data, as suggested by this Reviewer. Specifically:

Data in Ref 5 and 6 are unequivocal in demonstrating that ablation of dystrophin (Ref 6) and dystroglycan (which evokes loss of dystrophin in sarcolemma, Ref 5) were not associated with the typical dystrophic muscle degeneration-regeneration. The explanation that lack of abnormalities may be due to the analysis made in tibialis anterior is contradicted by all the studies in mdx mice, in which TA have been the mostly analysed muscle, very much because it shows typical dystrophic abnormalities.

Ref 7 has not been disproved by Spinazzola et al., as the authors themselves concluded in the abstract: “Although we did not identify an effective drug, we believe our data is of value to the DMD research community for future studies, and there is evidence that suggests serotonin modulation may still be a viable treatment strategy with further investigation”. Moreover, this was just one example we gave for treatments that can improve dystrophic symptoms in the absence of dystrophin.

Ref 8 can be used to illustrate that “myofibers can function entirely without dystrophin” as two different species (dogs and zebrafish) completely lacking dystrophin but having Jagged 1 transcription factor upregulation have functional muscle and normal lifespan. Nevertheless, we have modified this statement to say “myofibers can function without dystrophin” to avoid an overstatement.

Lane 426 statement is also correct. Contrary to what is currently believed, these two papers demonstrate that DAPs can exist in the sarcolemma without the anchoring dystrophin. In Ref 6, the Figure 8 illustrates some minor (in fact, indistinguishable) downregulation but not the loss of dystroglycan and sarcoglycan, while Ref 5 even contradicts the current dogma on dystrophin anchoring the DAPC, as it shows the opposite effect – sarcolemmal dystrophin loss being triggered by the depletion of dystroglycan. And in both studies, there was no muscle damage typical for DMD, including no increased necrosis or fragility of the sarcolemma.

Lanes 436-441 merely reinforce the message that the DMD pathology does not start with the loss of dystrophin in differentiated myofibers but it is a complex disease and that this complexity must be considered when trying to find an effective treatment.

Authors should indicate the source of myoblasts from DMD subjects (biopsies or cell biobank) and the methods used for their purification. Furthermore, they must provide ethic statement for the use of human samples.

This information was included in our paper (Mournetas et al., 2021), which we cited in the manuscript. Nevertheless, it was now included in full in the revised text.

Reviewer #2 (Recommendations for the authors):The authors extensively discussed many potentially affected molecular pathways in the DMD myoblasts as reasons for the observed alteration in proliferation, migration, and differentiation. A mechanistic study of one or more pathways will significantly increase the manuscript's merit.

As we have explained above, in this manuscript we not only demonstrated altered molecular pathways but we proved experimentally that these pathways are functionally altered. A further mechanistic study would certainly extend this important finding that dystrophic myoblasts are affected by the disease. To understand the mechanistic link between the loss of full-length dystrophin and myoblast defects, we are investigating whether mini-dystrophin re-expression in myoblasts can alleviate the dystrophic abnormalities. We found that it indeed normalized the debilitating purinergic phenotype occurring in these cells (which is another DMD manifestation discovered in myoblasts). However, completion of such a study is a lengthy and expensive project, which is beyond reasonable feasibility for this manuscript under review.